# Counterfactual harm

**Jonathan G. Richens**\*
DeepMind

**Rory Beard**
Comind

**Daniel H. Thompson**
Babylon

## Abstract

To act safely and ethically in the real world, agents must be able to reason about harm and avoid harmful actions. However, to date there is no statistical method for measuring harm and factoring it into algorithmic decisions. In this paper we propose the first formal definition of harm and benefit using causal models. We show that any factual definition of harm is incapable of identifying harmful actions in certain scenarios, and show that standard machine learning algorithms that cannot perform counterfactual reasoning are guaranteed to pursue harmful policies following distributional shifts. We use our definition of harm to devise a framework for harm-averse decision making using counterfactual objective functions. We demonstrate this framework on the problem of identifying optimal drug doses using a dose-response model learned from randomized control trial data. We find that the standard method of selecting doses using treatment effects results in unnecessarily harmful doses, while our counterfactual approach identifies doses that are significantly less harmful without sacrificing efficacy.

## 1 Introduction

Machine learning algorithms are increasingly used for making highly consequential decisions in areas such as medicine [1, 2, 3], criminal justice [4, 5, 6], finance [7, 8, 9], and autonomous vehicles [10]. These algorithms optimize for outcomes such as survival time, recidivism rate, and financial cost. However, these are purely factual objectives, depending only on *what* outcome occurs. On the other hand, human decision making incorporates preferences that depend not only on the outcome, but *why* it occurred.

For example, consider two treatments for a disease which, when left untreated, has a $50\%$ mortality rate. Treatment 1 has a $60\%$ chance of curing a patient, and a $40\%$ chance of having no effect, with the disease progressing as if untreated. Treatment 2 has an $80\%$ chance of curing a patient and a $20\%$ chance of killing them. Treatments 1 and 2 have identical recovery rates, hence any agent that selects actions based solely on the factual outcome statistics (e.g. the effect of treatment on the treated [11]) will not be able to distinguish between these two treatments. However, doctors systematically favour treatment 1 as it achieves the same recovery rate but never harms the patient—there are no patients that would have survived had they not been treated. On the other hand, doctors who administer treatment 2 risk harming their patients—there are patients who die following treatment who would have lived had they not been treated. While treatments 1 and 2 have the same mortality rates, the resulting deaths have different causes (doctor or disease) which have different ethical implications.

Determining what would have happened to a patient if the doctor hadn't treated them is a counterfactual inference, which cannot be identified from outcome statistics alone but requires knowledge of the functional causal relation between actions and outcomes [12]. However, standard machine

---

\*jonrichens@deepmind.com

learning algorithms are limited to making factual inferences—e.g. learning correlations between actions and outcomes. How then can we express these ethical preferences as objective functions and ultimately incorporate them into machine learning algorithms? This question has some urgency, given that machine learning algorithms currently under development or in deployment will be unable to reason about harm and factor it into their decisions.

The concept of harm is foundational to many ethical codes and principles; from the Hippocratic oath [13] to environmental policy [14], and from the foundations of classical liberalism [15], to Asimov's laws of robotics [16]. Despite this ubiquity, there is no formal statistical definition of harm. We address this by translating the predominant account of harm [17, 18] into a statistical measure—*counterfactual harm*. Our definition of harm enables us for the first time to precisely answer the following basic questions,

**Question 1** Did the actions of an agent cause harm, and if so, how much?

**Question 2** How much harm can we expect an action to cause prior to taking it?

**Question 3** How can we identify actions that balance the expected harms and benefits?

In section 2 we introduce concepts from causality, ethics, and expected utility theory used to derive our results. In section 3 we present our definitions of harm and benefit, and in section 4 we describe how harm-aversion can be factored into algorithmic decisions. In section 5 we prove that algorithms based on standard statistical learning theory are incapable of reasoning about harm and are guaranteed to pursue harmful policies following certain distributional shifts, while our approach using counterfactual objective functions overcomes these problems. Finally, in section 6 we apply our methods to an interpretable machine learning model for determining optimal drug doses, and compare this to standard algorithms that maximize treatment effects.

## 2 Prelimiaries

In this section we introduce some basic principles from causal modelling, expected utility theory and ethics that we use to derive our results.

### 2.1 Structural causal models

We provide a brief overview of structural causal models (SCMs) (see Chapter 7 of [19] and [20] for reviews). SCMs represent observable variables $X^i$ as vertices of a directed acyclic graph (DAG), with directed edges representing causal relations between $X^i$ and their direct causes (parents), which includes observable (endogenous) parents $\mathrm{pa}_i$ and a single exogenous noise variable $E^i$. The value of each variable is assigned by a deterministic function $f_i$ of these parents.

**Definition 1** (Structural Causal Model (SCM)). *A structural causal model $\mathcal{M} = \langle \mathcal{G}, E, X, F, P \rangle$ specifies:*

1. *a set of observable (endogenous) random variables $X = \{X^1, \ldots, X^N\}$,*

2. *a set of mutually independent noise (exogenous) variables $E = \{E^1, \ldots, E^N\}$ with joint distribution $P(E = e) = \prod_{i=1}^N P(E^i = e^i)$ (assuming causal sufficiency [19])*

3. *a directed acyclic graph $\mathcal{G}$, whose nodes are the variables $E \cup X$ with directed edges from $pa_i$, $E^i$ to $X^i$, where $pa_i \subseteq X$ denotes the endogenous parents of $X^i$.*

4. *a set of functions (mechanisms) $F = \{f_1, \ldots, f_N\}$, where the collection $F$ forms a mapping from $E$ to $X$, $x^i := f_i(pa_i, e^i)$, for $i = 1, \ldots, N$,*

By recursively applying condition 4 every observable variable can be expressed as a function of the noise variables alone, $X^i(e) = x^i$ where $E = e$ is the joint state of the noise variables, and the distribution over unobserved noise variables induces a distribution over the observable variables,

$$P(X = x) = \int_{e:X(e)=x} P(E = e) \tag{1}$$

where $X = x$ is the joint state over $X$.

The power of SCMs is that they specify not only the joint distribution $P(X = x)$ but also the distribution of $X$ under all interventions, including incompatible interventions (counterfactuals). Interventions describe an external agent changing the underlying causal mechanisms $F$. For example, $do(X^1 = x)$ denotes intervening to force $X^1 = x$ regardless of the state of its parents, replacing $x^1 := f_1(pa_1, e^1) \rightarrow x^1 := x$. The variables in this post-intervention SCM are denoted $X^i_x$ or $X^i_{X^1=x}$ and their interventional distribution is given by (1) under the updated mechanisms.

These interventional distributions quantify the effects of actions and are used to inform decisions. For example, the conditional average treatment effect (CATE) [21],

$$\text{CATE}(x) = \mathbb{E}[Y_{T=1}|x] - \mathbb{E}[Y_{T=0}|x] \tag{2}$$

measures the expected value of outcome $Y$ for treatment $T = 1$ compared to control $T = 0$ conditional on $X$, and is used in decision making from precision medicine to economics [22, 23].

Counterfactual distributions determine the probability that other outcomes would have occurred had some precondition been different. For example, for $X, Y \in \{T, F\}$, the causal effect $P(Y_{X=T} = T)$ is insufficient to determine if $do(X = T)$ was a necessary cause of $Y = T$ [24]. Instead this is captured by the counterfactual probability of necessity [25],

$$\text{PN} = P(Y_{X=F} = F | X = T, Y = T) \tag{3}$$

which is the probability that $Y$ would be false if $X$ had been false (the counterfactual proposition), given that $X$ and $Y$ were both true (the factual data). For example, if PN = 0 then $Y = T$ would have occured regardless of $X = T$, whereas if PN = 1 then $Y = T$ could not have occurred without $X = T$. Equation (3) involves the joint distribution over incompatible states of $Y$ under different interventions, $Y = T$ and $Y_{X=F} = F$, which cannot be jointly measured and hence (3) cannot be identified from data alone. However, (3) can be calculated from the SCM using (1) for the combined factual and counterfactual propositions,

$$P(Y_{X=F} = F, X = T, Y = T) = \int_{\substack{Y_{X=F}(e)=F \\ Y(e)=T, X(e)=T}} P(E = e) \tag{4}$$

This ability to ascribe causal explanations for data has made SCMs and counterfactual inference a key ingredient in statistical definitions of blame, intent and responsibility [26, 27, 28], and has seen applications in AI research ranging from explainability [29, 30], safety [31, 32], and fairness [33], to reinforcement learning [34, 35] and medical diagnosis [36].

Our definitions and theoretical results are derived within the SCM framework as is standard in causality research [19]. However, there is a growing body of work developing deep learning models that are capable of supporting counterfactual reasoning in complex domains including medical imaging [37], visual question answering [38], and vision-and-language navigation in robotics [39] and text generation [40]. We discuss these approaches in relation to our work in Appendix L.1.

## 2.2 The counterfactual comparative account of harm

The concept of harm is deeply embedded in ethical principles, codes and law. One famous example is the bioethical principle "first, do no harm" [41], asserting that the moral responsibility for doctors to benefit patients is superseded by their responsibility not to harm them [42]. Another example is John Stuart Mill's harm principle which forms the basis of classical liberalism [15] and inspired the "zeroth law" of Asimov's fictional robot governed society, which states that "A robot may not harm humanity or, by inaction, allow humanity to come to harm" [16].

Despite these attempts to codify harm into rules for governing human and AI behaviour, it is not immediately clear what we mean when we talk about harm. This can lead to confusion—for example, does "allow humanity to come to harm" describe an agent causing harm by inaction [43] or failing to benefit? The meaning and measure of harm also has important practical ramifications. For example, establishing negligence in tort law requires establishing that significant harm has occurred [44].

This has motivated work in philosophy, ethics and law to rigorously define harm, with the most widely accepted definition being the *counterfactual comparative account* (CCA) [17, 45, 46],

**Definition 2** (CCA). *The counterfactual comparative account of harm and benefit states,*

> *"An event $e$ or action $a$ harms (benefits) a person overall if and only if she would have been on balance better (worse) off if $e$ had not occurred, or $a$ had not been performed."*

The CCA quantifies harm by comparing how well off a person is (their factual outcome) compared to how well off they would have been (their counterfactual outcome) had the agent not acted. Generating these counterfactuals necessitates a causal model of the agents actions and their consequences, and we determine how "better (worse) off" the person is using an expected utility analysis [47]. While the wording of the CCA suggests that we always compare to the counterfactual world where the agent takes no action, a more natural and general approach is to measure harm compared to a specific default action or policy. For example, when measuring the harm caused by a medicine we may intuitively want to compare to the outcomes that would have occurred with a placebo, and for the harm caused by negligent medical decisions the natural choice is to compare to a standard clinical decision policy. To this end, we measure harm compared to a default action (or policy), see Appendix D) for further discussion.

While the CCA is the predominant definition of harm and forms the basis of our results, alternative definitions have been proposed [48, 49], motivated by scenarios where the CCA appears to give counter-intuitive results [43]. Surprisingly, we find these problematic scenarios are identical to those raised in the study of actual causality [24], and can be resolved with a formal causal analysis. To our knowledge, this connection between the harm literature and actual causality has not been noted before, and we discuss these alternative definitions in Appendix C and how the arguments surrounding the CCA can be resolved in Appendix B in the hope this will stimulate discussion between these two fields.

### 2.3 Decision theoretic setup

In this section we present our setup for calculating the CCA (Definition 2). The CCA describes a person (referred to as the *user*) being made better or worse off due to the actions of an agent. We focus on the simplest case involving single agents performing single actions, though ultimately our definitions can be extended to multi-agent or sequential decision problems (as discussed in Appendix L.3). For causal modelling in these scenarios we refer the reader to [32, 50, 51].

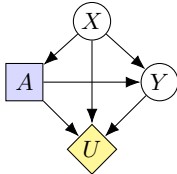

Figure 1: SCM $\mathcal{M}$ depicting the agent's action $A$, context $X$, outcomes $Y$ and utility function $U$.

We describe the environment with a structural causal model $\mathcal{M}$ which specifies the distribution over environment states $W$ as endogenous variables, $P(w; \mathcal{M})$. For single actions the environment variables can be divided into three disjoint sets $W = A \cup X \cup Y$ (Figure 1 a)); the agent's action $A$, outcome variables $Y$ that are descendants of $A$ and so can be influenced by the agent's action, and context variables $X$ that are non-descendants of nor in $A$. For simplicity we assume no unobserved confounders (causal sufficiency [19]) in the derivation of our theoretical results. In the following we assume knowledge of the SCM, and in Appendix L.1 we discuss this assumption and describe how tight upper bounds on the harm can be determined when the SCM is unknown.

The user's preferences over environment states are specified by their utility function $U(w) = U(a, x, y)$, and for a given action the user's expected utility is,

$$\mathbb{E}[U|a, x] = \int_y P(y|a, x)U(a, x, y) \tag{5}$$

In sections 3-4 we focus on the case where the agent acts to maximize the user's expected utility, with optimal actions $a_{\max} = \arg\max_a \mathbb{E}[U|a, x]$. While this is similar to the standard setup in expected utility theory [47, 52, 53], note that $U$ describes the preferences of the user, and in general the agent

can pursue any objective and could be entirely unaware of the user's preferences. In section 5 we extend our analysis to agents pursuing arbitrary objectives.

**Example 1: conditional average treatment effects.** Returning to our example from section 1, consider an agent choosing between a placebo $A = 0$, or treatments 1 and 2. $Y = 1$ indicates recovery and $Y = 0$ mortality. In the trial, harm is measured by comparison to 'no treatment' $\bar{a} = 0$. $X = x$ describes any variables that can potentially influence the agent's choice of intervention and/or $Y$, e.g. the patient's medical history.

Consider patients (users) whose preferences are fully determined by survival, e.g. $U(a, x, y) = y$. The expected utility is $\mathbb{E}[U|a, x] = \sum_y P(y|a, x)U(a, x, y) = \sum_y yP(y_a|x) = \mathbb{E}[Y_a|x]$, where in the last line we have use the fact that there are no confounders between $A$ and $Y$. The agent's policy is therefore to maximize the treatment effect $a_{\max} = \arg\max_a \mathbb{E}[Y_a|x]$, which is equivalent to $a_{\max} = \arg\max_a\{\mathbb{E}[Y_a|x] - \mathbb{E}[Y_0|x]\}$ i.e. maximizing the CATE (2)—a standard objective for identifying optimal treatments [54, 55]. In Appendix G we derive an SCM model for the treatments described in the introduction, and show that $\mathbb{E}[Y_{A=1}] = \mathbb{E}[Y_{A=2}] = 0.8$, i.e. the expected utility and CATE is the same for treatments 1 and 2.

## 3    Counterfactual harm

We now present our definitions for harm and benefit based on Definition 2. For the examples described in this paper we focus on simple environments with single outcome variables, for which harm can be determined by comparing the factual utility received by the user to the utility they would have received had the agent taken some default action $A = \bar{a}$ (Definition 3). In Appendix D we describe how this default action can be determined. Some more complicated situations call for a path-dependent measure of harm, which we provide in Appendix A along with examples in Appendices B-D. In the following, counterfactual states are identified with an $*$, e.g. $Z = z^*$. First we define the harm caused by an action given we observe the outcome (Question 1).

**Definition 3** (Counterfactual harm & benefit). *The harm caused by action $A = a$ given context $X = x$ and outcome $Y = y$ compared to the default action $A = \bar{a}$ is,*

$$h(a, x, y; \mathcal{M}) = \int_{y^*} P(Y_{\bar{a}} = y^*|a, x, y; \mathcal{M})\max\{0, U(\bar{a}, x, y^*) - U(a, x, y)\} \tag{6}$$

*and the expected benefit is*

$$b(a, x, y; \mathcal{M}) = \int_{y^*} P(Y_{\bar{a}} = y^*, |a, x, y; \mathcal{M})\max\{0, U(a, x, y) - U(\bar{a}, x, y^*)\} \tag{7}$$

*where $\mathcal{M}$ is the SCM describing the environment and $U$ is the user's utility.*

The counterfactual harm (benefit) is the expected increase (decrease) in utility had the agent taken the default action $A = \bar{a}$, given they took action $A = a$ in context $X = x$ resulting in outcome $Y = y$. Taking the max in (6) and (7) ensures that we only include counterfactual outcomes where the utility is higher (lower) in the counterfactual world. Note that $X_{\bar{a}} = X$ as $X$ is not a descendant of $A$, so the factual and counterfactual contexts are identical [56]. While we focus on the standard decision theoretic analysis of actions, it is trivial to extend our analysis to naturally occurring events that are not determined by an agent (e.g. by identify $A$ with an event rather than an action).

To determine the expected harm or benefit of an action prior to taking it (Question 2) we calculate the expected value of (6) and (7) over the outcome distribution, e.g. $\mathbb{E}[h|a, x; \mathcal{M}] = \int_y P(y|a, x; \mathcal{M})h(a, x, y; \mathcal{M})$. Note these form a counterfactual decomposition of the expected utility.

**Theorem 1** (harm-benefit trade-off). *The difference in expected utility for action $A = a$ and the default action $A = \bar{a}$ is given by,*

$$\mathbb{E}[U|a, x] - \mathbb{E}[U|\bar{a}, x] = \mathbb{E}[b|a, x; \mathcal{M}] - \mathbb{E}[h|a, x; \mathcal{M}] \tag{8}$$

(Proof: see Appendix F).

**Example 2:** Returning to Example 1, the expected counterfactual harm for default action $\bar{a} = 0$ is, $\mathbb{E}[h|a; \mathcal{M}] = \sum_{y^*, y} P(Y_{\bar{a}} = y^*, y|a; \mathcal{M}) \max\{0, y^* - y\} = P(Y_0 = 1, Y_a = 0; \mathcal{M})$ where we have used $U(a, y) = y$ and the fact that $Y_{\bar{a}} \perp A$ and there are no latent confounders to give $P(Y_0 = y^*, Y = y|a) = P(Y_0 = y^*, Y_a = y)$. The harm is therefore the probability that a patient dies following treatment $A = a$ and would have recovered had they received no treatment. In Appendix G we show that $P(Y_0 = 1, Y_1 = 0) = 0$ (treatment 1 causes zero harm) and $P(Y_0 = 1, Y_2 = 0) = 0.1$ (treatment 2 causes non-zero harm), reflecting our intuition that treatment 2 is more harmful than treatment 1, despite having the same causal effect / CATE. Hence, the counterfactual harm allows us to differentiate between these two treatments.

## 4 Harm in decision making

Now we have expressions for the expected harm and benefit, how can we incorporate these into the agent's decisions? Consider two actions $a, a'$ such that $\mathbb{E}[b|a', x; \mathcal{M}] = \mathbb{E}[b|a, x; \mathcal{M}] + K$ and $\mathbb{E}[h|a', x; \mathcal{M}] = \mathbb{E}[h|a, x; \mathcal{M}] + K$ where $K \in \mathbb{R}$. By Theorem 1 they must have the same expected utility, as $\mathbb{E}[U|a', x] = \mathbb{E}[b|a', x; \mathcal{M}] - \mathbb{E}[h|a', x; \mathcal{M}] + \mathbb{E}[U|\bar{a}, x] = \mathbb{E}[b|a, x; \mathcal{M}] + K - \mathbb{E}[h|a, x; \mathcal{M}] - K + \mathbb{E}[U|\bar{a}, x] = \mathbb{E}[U|a, x]$. Therefore expected utility maximizers are indifferent to harm, and will are willing to increase harm for an equal increase in benefit.

We say that an agent is *harm averse* if they are willing to risk causing harm only if they expect a comparatively greater benefit. Theorem 1 suggests a simple way to overcome harm indifference in the expected utility by assigning a higher weight to the harm component of the decomposition (8). If instead we maximize an adjusted expected utility $\mathbb{E}[V|a, x; \mathcal{M}] := \mathbb{E}[U|x] + \mathbb{E}[b|a, x; \mathcal{M}] - (1 + \lambda)\mathbb{E}[h|a, x; \mathcal{M}]$ where $\lambda \in \mathbb{R}$, then if $\mathbb{E}[V|a, x; \mathcal{M}] = \mathbb{E}[V|a', x; \mathcal{M}]$ and $\mathbb{E}[h|a', x; \mathcal{M}] = \mathbb{E}[h|a, x; \mathcal{M}] + K$ then $\mathbb{E}[b|a', x; \mathcal{M}] = \mathbb{E}[b|a, x; \mathcal{M}] + (1 + \lambda)K$, and the agent is willing to trade harm for benefit at a $1 : (1 + \lambda)$ ratio. Taking inspiration from risk aversion [57], we refer to $\lambda$ as the agent's *harm aversion*. To achieve the desired trade-off we replace the utility function with the harm-penalized utility (HPU),

**Definition 4** (Harm penalized utility (HPU)). *For utility $U$ and an environment described by SCM $\mathcal{M}$, the harm-penalized utility (HPU) is,*

$$V(a, x, y; \mathcal{M}) = U(a, x, y) - \lambda h(a, x, y; \mathcal{M}) \tag{9}$$

*where $h(a, x, y; \mathcal{M})$ is the harm (Definition 3) and $\lambda$ is the harm aversion.*

For $\lambda = 0$ the agent is *harm indifferent* and optimizes the standard expected utility. For $\lambda > 0$ the agent is *harm averse*, and is willing to reduce the expected utility to achieve a greater reduction to the expected harm. For $\lambda < 0$ the agent is *harm seeking*, and is willing to reduce the expected utility in order to increase the expected harm.

Maximizing the expected HPU allows agents to choose actions that balance the expected benefits and harms (Question 2 section 1). We refer to the expected HPU (9) as a *counterfactual objective function* as $V(a, x, y; \mathcal{M})$ is a counterfactual expectation that cannot be evaluated without knowledge of $\mathcal{M}$. We refer to the expected utility as a *factual objective function*, as it can be evaluated on data alone without knowledge of $\mathcal{M}$. As we show in the following example, harm-aversion can lead to very different behaviours compared to other factual approaches to safety such as risk aversion [57].

**Example 3: assistant paradox.** Alice invests \$80 in the stock market, and expects a normally distributed return $Y \sim \mathcal{N}(\mu, \sigma^2)$ with $\mu = \sigma = \$100$. She asks her new AI assistant to manage her investment, and after a lengthy analysis the AI identifies three possible actions;

1. Multiply the investment by $0 \le K \le 20$, resulting in a return $Y \to KY$,
2. Perform a clever trade that Alice was unaware of, increasing her return by $+\$10$ with certainty, $Y \to Y + 10$,
3. Cancel Alice's investment, returning \$80.

Consider 3 agents,

1. Agent 1 maximizes expected return, choosing $a = \arg\max_a \mathbb{E}[Y|a]$,
2. Agent 2 is risk-averse, in the sense of [57], choosing $a = \arg\max_a \{\mathbb{E}[Y|a] - \lambda \text{Var}[Y|a]\}$,

3. Agent 3 is harm-averse, choosing $a = \arg\max_a \{\mathbb{E}[Y|a] - \lambda\mathbb{E}[h|a; \mathcal{M}]\}$.

In Appendix H we derive the optimal policies for Agents 1-3, and calculate the harm with respect to the default action where the agent leaves Alice's investment unchanged (i.e. action 1, $K = 1$). Agent 1 chooses action 1 and $K = 20$, maximising Alice's return but also maximizing the expected harm—e.g. if Alice would have lost \$100 she will now lose \$2000. The standard approach in portfolio theory [58] is to use agent 2 with an appropriate risk aversion $\lambda$. However, agent 2 never chooses action 2 for any $\lambda$, despite action 2 always increasing Alice's return (causing zero harm and non-zero benefit). Agent 2 will reduce or even cancel Alice's investment (for $\lambda > 0.0032$) rather than take action 2. This is because the risk-averse objective is factual, and cannot differentiate between losses caused by the agent's actions from those that would have occurred regardless of the agent's actions (i.e. due to exogenous fluctuations in return). Consequently, agent 2 treats all outcomes as being caused by its actions, and is willing to harm Alice by reducing or even cancelling her investment in order to minimize her risk. On the other hand agent 3 never reduces Alice's expected return, choosing either action 1 or action 2 depending on the degree of harm aversion.

The surprising behaviour of agent 2 highlights that factual objectives (e.g. risk aversion) cannot capture preferences that depend not only on the outcome but on its cause. Alice may assign a different weight to losses that are caused by her assistant's actions rather than by exogenous market fluctuations or her own actions—e.g. if she places a bet that would have made a return but was overridden by her assistant. Consequently she may prefer action 2 over action 1 or action 3 with $K < 1$. While this can be achieved using a factual objective function that favours action 2 a priori (assuming Alice is aware of this action), next we show this approach fails to generalize and actually leads to harmful policies.

## 5 Counterfactual reasoning is necessary for harm aversion

In the previous sections we saw examples where expected utility maximizers harm the user. However, these examples describe specific environments and utility functions, and assume the agent maximizes the expected utility instead of some other (potentially safer) objective. In this section we consider users with arbitrary utility functions and agents with arbitrary objectives to answer; when is maximizing the expected utility harmful? And are there other factual objective functions that can overcome this?

Factual objective functions can be expressed as the expected value of a real-valued function $J(a, x, y)$. Examples include the expected utility, cost functions [59], and the cumulative discounted reward [60]. Once $J$ is specified the optimal actions in environment $\mathcal{M}$ are given by,

$$a_{\max} = \arg\max_a \mathbb{E}[J|a, x] = \arg\max_a \int_y P(y|a, x; \mathcal{M})J(a, x, y) \tag{10}$$

Note that (10) can be evaluated from data alone (i.e. from samples of the joint distribution of $A$, $X$ and $Y$) without knowledge of $\mathcal{M}$. In the following we relax our requirement that agents have some fixed harm aversion $\lambda$, requiring only that they don't take strictly harmful actions.

**Definition 5** (Harmful actions & policies). *An action $A = a$ is strictly harmful in context $X = x$ and environment $\mathcal{M}$ if there is $a' \neq a$ such that $\mathbb{E}[U|a', x] \geq E[U|a, x]$ and $\mathbb{E}[h|a', x; \mathcal{M}] < \mathbb{E}[h|a, x; \mathcal{M}]$. A policy is strictly harmful if it assigns a non-zero probability to a strcitly harmful action.*

**Definition 6** (Harmful objectives). *An objective $\mathbb{E}[J|a, x]$ is strictly harmful in environment $\mathcal{M}$ if there is a policy that maximizes the expected value of $J$ and is strictly harmful.*

Note that strictly harmful actions violate any degree of harm aversion $\lambda > 0$, as the agent is willing to choose actions that are strictly more harmful for no additional benefit. In examples 1 and 2, treatment 2 is a strictly harmful action and the CATE is a strictly harmful objective for this decision task. First, we show that the expected HPU is not a strictly harmful objective in any environment,

**Theorem 2.** *For any utility functions $U$, environment $\mathcal{M}$ and default action $A = \bar{a}$ the expected HPU is not a strictly harmful objective for any $\lambda > 0$. (Proof: see Appendix K).*

It is vital that agents continue to pursue safe objectives following changes to the environment (distributional shifts). At the very least, agents should be able to re-train in the shifted environment without needing to tweak their objective functions. In SCMs distributional shifts are represented as interventions that change the exogenous noise distribution and/or the underlying causal mechanisms [61, 62]. We focus on distributional shifts that change the outcome distribution $P(y|a, x; \mathcal{M})$ alone.

**Definition 7** (Outcome distributional shift). *For an environment described by SCM $\mathcal{M}$, $\mathcal{M} \rightarrow \mathcal{M}'$ is a shift in the outcome distribution if the exogenous noise distribution for $Y$ changes $P(e^Y; \mathcal{M}') \neq P(e^Y; \mathcal{M})$ and/or the causal mechanism changes $f_Y'(a, x, e^Y) \neq f_Y(a, x, e^Y)$.*

By Theorem 2 the HPU is never strictly harmful following any distributional shift, as it is not strictly harmful in any $\mathcal{M}'$. On the other hand, we find that maximizing almost any factual utility function will result in harmful policies under distributional shifts. In the following we assume some degree of outcome dependence in the user's utility function.

**Definition 8** (Outcome dependence). *$U(a, x, y)$ is outcome dependent for a set of actions $C = \{a_1, \ldots, a_N\}$ in context $X = x$ if $\forall a_i, a_j \in C, i \neq j, \max_y U(a_i, x, y) > \min_y U(a_j, x, y)$.*

If there is no outcome dependence then the optimal action is independent of the outcome distribution $P(y|a, x)$, and no learning is required to determine the optimal policy. Typically we are interested in tasks that require some degree of learning and hence outcome dependence.

**Theorem 3.** *If for any context $X = x$ the user's utility function is outcome dependent for the default action in that context $\bar{a}(x)$ and another action $a \neq \bar{a}(x)$, then there is an outcome distributional shift such that $U$ is strictly harmful in the shifted environment.* (Proof: see Appendix K).

To understand the implications of Theorem 3 we can return to examples 1 & 2 and no longer assume a specific utility function or outcome statistics. Theorem 3 implies that for an agent maximizing the user's utility, there is always a distributional shift such that the agent will cause unnecessary harm to patient in the shifted environment, compared to not treating them ($\bar{a} = 0$) or any other default action. Unlike in examples 1 & 2 we allow the patient's utility function to depend on the treatment choice, and for treatments to vary in effectiveness. The only assumption we make is that whatever the user's utility function is, it is not true that $U(A = a, x, Y = 0) > U(A = 0, x, Y = 1) \, \forall \, a, x$. If this were true then the patient's preferences are dominated by the agents action choice alone, i.e. the user would always prefer to die so long as they are treated, rather than not being treated and surviving.

While robust harm aversion is not possible for expected utility maximizers in general, it seems likely at first that we can train robustly harm-averse agents using other learning objectives. One possibility would be to use human interactions and demonstrations [63, 64, 65] that *implicitly* encode harm aversion, with $J(a, x, y)$ representing user feedback or some learned (factual) reward model. In examples 1 & 2 users with utility $U(a, x, y) = y$ could be altered to assign a higher cost to deaths following treatment 2 compared to treatment 1, $J(A = 2, Y = 0) < J(A = 1, Y = 0)$, with $J$ implicitly encoding the harm caused by treatment 2. If there is some choice of $J$ that implicitly encodes this harm-aversion in all shifted environments, this would allow robust harm-averse agents to be trained using standard statistical learning techniques without needing to learn $\mathcal{M}$ or perform counterfactual inference. However, we now show this is not possible in general.

**Theorem 4.** *If for any context $X = x$ the user's utility function is outcome dependent for the default action in that context $\bar{a}(x)$ and two other actions $a_1, a_2 \neq \bar{a}(x)$, then for any factual objective function $J$ there is an outcome distributional shift such that $J$ is strictly harmful in the shifted environment.* (Proof: see Appendix K).

Theorem 4 has profound consequences for the possibility of training safe agents with standard machine learning algorithms. It implies that agents optimizing factual objective functions may appear safe in their training environments but are guaranteed to pursue harmful policies (with respect to Definition 2) following certain distributional shifts, even if they are allowed to re-train. This is true regardless of the (factual) objective function we choose for the agent, and applies to any user whose utility function has some basic degree of outcome dependence. This is particularly concerning as all standard machine learning algorithms use factual objective functions. In section 6 we give a concrete example of a reasonable factual objective function and a distributional shift that renders it harmful.

**Example 4:** An AI assistant manages Alice's health including medical treatments, clinical testing and lifestyle interventions. These actions have a causal effect on Alice's health outcomes—e.g. disease progression, severity of symptoms, and medical costs, and Alice's preferences depend on these outcomes and not just on the action chosen by the agent (outcome dependence). The agent maximizes a reward function over these health states and actions (factual objective), including feedback from Alice and clinicians in-the-loop which penalize harmful actions (implicit harm aversion) by comparing to standardised treatment rules (default actions). The agent appears to be safe and is deployed at scale. However, by Theorem 4 there exists a distributionally shifted environment such that the agent's

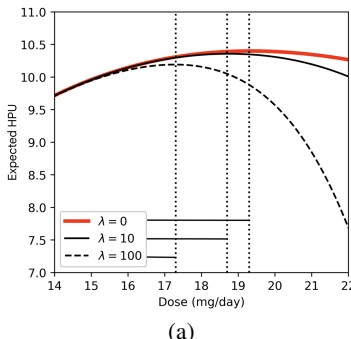
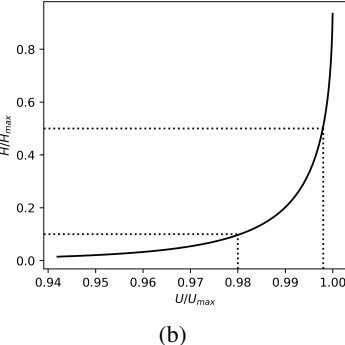

|       |       |
| :---: | :---: |
| (a)   | (b)   |

Figure 2: a) Expected HPU (Def 4) for the dose-response model (11) v.s. dosage of for $\lambda = 0$, $\lambda = 10$ and 100. Dotted lines show optimal doses. b) $\mathbb{E}[h|a]/\mathbb{E}[h|a_{\max}]$ v.s. $\mathbb{E}[U|a]/\mathbb{E}[U|a_{\max}]$ for $a \leq a_{\max}$. Dotted lines show doses reducing harm by 50% and 90% and utility by 0.2% and 2.0% respectively.

optimal policy is strictly harmful (Definition 5), in that it chooses needlessly harmful treatments for the patient, and this is true even if we train to convergence in the shifted environment.

## 6 Illustration: Dose-response models

How does harm aversion affect behaviour and performance in real-world tasks? And how can reasonable objective functions result in harmful policies following distributional shifts? In this section we apply the methods developed in section 4 to the task of determining optimal treatments. This problem has received much attention in recent years following advances in causal machine learning techniques [66] and growing commercial interests including recommendation [67], personalized medicine [55], epidemiology and econometrics [68] to name a few.

While harm is important for decision making in many of these areas, existing approaches reduce the problem to identifying treatment effects—e.g. maximizing the CATE (2)—which is indifferent to harm as shown in sections 3 and 5. This could result in needlessly harmful decisions being made, perhaps overlooking equally effective decisions that are far less harmful.

To demonstrate this we use a model for determining the effectiveness of the drug Aripiprazole learned from a meta-analysis of randomised control trial data [69]. This dose-response model predicts the reduction in symptom severity $Y$ following treatment with a dose $A$ (mg/day) of Aripiprazole using a generalized additive model (GAM) [70]—an interpretable machine learning model commonly used for dose-response analysis [71],

$$y = f(a, \theta_1, \theta_2, \varepsilon) = \theta_1 a + \theta_2 f(a) + \varepsilon \tag{11}$$

where $\theta_i \sim \mathcal{N}(\hat{\theta}_i, V_i)$ are random effects parameters, $\varepsilon \sim \mathcal{N}(0, V_\varepsilon)$ is a noise parameter, and $f(a)$ is a cubic spline function (for details and parameters see Appendix J). We assume $U(a, y) = y$ and measure harm in comparison to the zero dose $\bar{a} = 0$. In Appendix J we derive a closed-form expression for the expected harm in GAMs.

We calculate the expected HPU (4) for $\lambda = 0, 10, 100$ (Figure 2 a)). In this context a relatively large $\lambda$ is appropriate to ensure a high ratio of expected benefit to harm required for medical interventions [72]. We find that the optimal dose is highly sensitive to $\lambda$, for example reducing from 19.3 mg/day to 17.3 mg/day for $\lambda = 100$. On the other hand, these lower-harm doses achieve a similar improvement in symptom severity compared to the optimal dose. Figure 2 b) shows the trade-off between expected utility and harm relative to the optimal dosage $a_{\mathrm{CATE}} = \arg\max_a \mathbb{E}[Y_a]$. For almost-optimal doses we observe a steep harm gradient, with small improvements in symptom severity requiring large increases in harm. For example, choosing a lower dose than $a_{\mathrm{CATE}}$ one can reduce harm by 90% while only reducing the treatment effect by 2%. This demonstrates that ignoring harm while maximizing expected utility will often result in extreme harm values (an example of Goodhart's law [73]).

Finally, we consider identifying safe doses using a reasonable factual objective function (risk-aversion) and describe how this can result in harm following distributional shifts. Consider the

risk-averse objective $\mathbb{E}[J|a] = \mathbb{E}[Y_a] - \beta\mathsf{Var}[Y_a]$. As $\mathsf{Var}[Y_a]$ increases with dose, this objective also selects lower harm doses in our example. However, consider a distributional shift where (11) gains an additional noise term, $y = f(a, \theta_1, \theta_2, \varepsilon, \eta) = \theta_1 a + \theta_2 f(a) + \varepsilon + \eta^Y(10 - 0.5a)$ where $\eta^Y \sim \mathcal{N}(0, 1)$, e.g. describing a shift where untreated patients ($A = 0$) have a high variation in outcome $Y$ which decreases as the dose increases. It is simple to check that, for $a = \arg\max_a \mathbb{E}[U|a]$ and $a' = \arg\max_a \mathbb{E}[J|a]$, $E[U|a] > E[U|a']$ and $E[h|a] < E[h|a'] \ \forall \ \beta > 0$ in this shifted environment. Hence risk averse agents are strictly harmful in the shifted environment (Definition 6).

## 7 Related work

In appendix L.3 we discuss two related works, path-specific counterfactual fairness [33, 74] and path-specific objectives [75], which similarly use counterfactual contrasts with a baseline to impose ethical constraints on objective functions.

Following the publication of our results an alternative causal definition of harm has been proposed [76, 77]. In the simplest settings with single actions and outcomes, this work defines an action as harmful in a deterministic environment if i) the utility is lower than a default value and ii) any other action the agent could have taken would have resulted in a higher utility. In Appendix E we describe this definition and present examples where it gives counter-intuitive results including attributing harm to unharmful actions. Furthermore, as described in [76] this definition is intractable except for simple causal models with low-dimensional variables, which limits its applicability to machine learning models (for example, this definition cannot be used for the dose-response model in section 6 which has a continuous action-outcome space).

## 8 Conclusion

One of the core assumptions of statistical learning theory is that human preferences can be expressed solely in terms of states of the world, while the underlying dynamical (i.e. causal) relations between these states can be ignored. This assumption allows goals to be expressed as factual objective functions and optimized on data without needing to learn causal models or perform counterfactual inference. According to this view, all behaviours we care about can be achieved using the first two levels of Pearl's hierarchy [78]. In this article we have argued against this view. We proposed the first statistical definition of harm and showed that agents optimizing factual objective functions are incapable of robustly avoiding harmful actions in general. We have demonstrated how our definition of harm can be used to meaningfully reduce the harms caused by machine learning algorithms, using the example of a simple model for determining optimal drug doses. The ability to avoid causing harm is a vital for the deployment of safe and ethical AI [79], and our results add weight to view that advanced AI systems will be severely limited if they are incapable of making causal and counterfactual inferences [80].

## 9 Acknowledgements

The authors would like to thank the NeurIPS reviewers for their details comments and to thank Tom Everitt, Ryan Carey, Albert Buchard, Sander Beckers, Hana Chockler and Joseph Halpern for their helpful discussions and comments on the manuscript.

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
