## Appendices

### A

In this appendix we present the general version of Definition 3 allowing harm and benefit to be measured along specific causal paths.

The path-specific counterfactual harm measures the harm caused by an action $A = a$ compared to a default action $A = \bar{a}$ when, rather than generating the counterfactual outcome by including all causal paths from $A = \bar{a}$ to outcome variables $Y$, we consider only the effect along certain paths $g$. This is somewhat analogous to the path specific causal effect [81], as we are using the $g$-specific intervention $A = \bar{a}$ on $Y$ in the counterfactual world relative to reference $A = a$ (the factual action).

**Definition 9** (Path-specific counterfactual harm & benefit). *Let $G$ be the DAG associated with model $\mathcal{M}$ and $g$ be the edge sub-graph of $G$ containing the paths we include in the harm analysis. The path specific harm caused by action $A = a$ compared to default action $A = \bar{a}$ is given by*

$$h_g(a, x, y; \mathcal{M}) = \int_{y^*} P(Y_{\bar{a}, \mathcal{M}_g} = y^* | a, x, y; \mathcal{M}) \max\{0, U(\bar{a}, x, y^*) - U(a, x, y)\} \tag{12}$$

$$= \int_{y^*, e} P(Y_{\bar{a}} = y^* | e; \mathcal{M}_g) P(e | a, x, y; \mathcal{M}) \max\{0, U(\bar{a}, x, y^*) - U(a, x, y)\} \tag{13}$$

*Where $Y_{\bar{a}, \mathcal{M}_g}$ is the counterfactual outcome $Y$ under intervention $do(A = \bar{a})$ in model $\mathcal{M}_g$ where $\mathcal{M}_g$ is formed from $\mathcal{M}$ by replacing the causal mechanisms for each variable $f^i(pa^i, e) \rightarrow f_g^i(pa^i(g)^*, e) = f^i(pa^i(g)^*, pa^i(\bar{g}), e)$, where $Pa^i(\bar{g})$ is the set of parents of $V^{(i)}$ that are not linked to $V^{(i)}$ in $g$ and $pa^i(\bar{g})$ is the factual state of those variables. $E = e$ is the joint state of the exogenous noise variables in $\mathcal{M}$. Likewise, the expected benefit is*

$$b_g(a, x, y; \mathcal{M}) = \int_{y^*} P(Y_{\bar{a}, \mathcal{M}_g} = y^* | a, x, y; \mathcal{M}) \max\{0, U(a, x, y) - U(\bar{a}, x, y^*)\} \tag{14}$$

Note that if we following the construction of $\mathcal{M}_g$ in [81] we get that $\mathcal{M}_g$ is formed from $\mathcal{M}$ by i) partitioning the parent set for each variable $V^{(i)}$ in $\mathcal{M}$ into $Pa^i = \{Pa^i(g), Pa^i(\bar{g})\}$ where $Pa^i(g)$ are the parents that are linked to $V^{(i)}$ in $g$ and $Pa^i(\bar{g})$ is the complimentary set, ii) replacing the mechanisms for each variable with $f^i(pa^i, e^i) \rightarrow f_g^i(pa^i, e^i) = f^i(pa^i(g)^*, pa^i(\bar{g}), e^i)$ where $pa^i(\bar{g})$ takes the value of $PA^i(\bar{g})_z$ in $\mathcal{M}$ where $A = z$ is the reference action. However, in (12) and (14) we condition on the state of all factual variables and assume no unobserved confounders, and the reference action is the factual action state. Therefore the state of $PA^i(\bar{g})_a$ in $\mathcal{M}$ is equal to the factual state of these variables, giving our simplified construction for $\mathcal{M}_g$.

We give examples of computing the path-specific harm in Appendices B-D. In these examples we typically focus on causal models with an action $A$, an outcome $Y$ and another mediating outcome $Z$ s.t. $A \rightarrow Z \rightarrow Y$ and $A \rightarrow Y$, and $U = U(Y)$. We refer to the path-specific harm where we restrict to $A \rightarrow Y$ as the 'direct harm', the path specific harm where we restrict to $A \rightarrow Z \rightarrow Y$ as the 'indirect harm', and the 'total harm' when we do not exclude any causal path.

### B

In this appendix we discuss the omission problem and pre-emption problem [43], and the preventing worse problem [82], and show how these can be resolved using our definition of counterfactual harm (Definition 3 and its path-specific variant Definition 9). We also discuss some alternative definitions of harm.

**Omission Problem:** Alice decides not to give Bob a set of golf clubs. Bob would be happy if Alice had given him the golf clubs. Therefore, according to the CCA, Alice's decision not to give Bob the clubs causes Bob harm. However, intuitively Alice has not harmed Bob, but merely failed to benefit him [43].

*Solution:* The omission problem relies on the judgement that Alice does not have a ethical obligation to provide Bob with golf clubs, therefore her choice not to do so does not constitute harm to Bob. In our definition of harm, this implies the obvious default action be that Alice not giving Bob clubs by default, i.e. the desired harm query is the harm caused by Alice's action compared to baseline where Alice does not give Bob club. To compute the harm we construct the model $\mathcal{M}$ comprising of two variables; Alice's action $A \in \{0,1\}$ where $A = 0$ indicates 'Bob not given clubs' and $A = 1$ 'Bob given clubs', and outcome $Y \in \{0,1\}$ where $Y = 1$ indicates 'Bob has clubs' and $Y = 0$ indicates 'Bob does not have clubs'. The default action $A = \bar{a}$ is $\bar{a} = 0$. The causal mechanism for $Y$ is $y = a$, i.e. Bob has clubs iff he is given them. Whatever utility function describes Bob's preferences, the action $A = 0$ causes no harm in this model (Lemma 3 Appendix K) as $P(Y_0 = y^*|A = 0, Y = y) = \delta(y^* - y)$ (factual $a$ and counterfactual $\bar{a}$ are identical) and for non-zero harm we require $y^* \neq y$.

Note there are other reasonable scenarios where Alice's actions would constitute harm. For example, if Alice was a clerk in a golf shop and Bob had pre-paid for a set of golf clubs, we could claim that 'the clerk Alice harmed Bob by not giving him golf clubs'. In this case, we would expect Alice to give Bob the clubs by default (she has a ethical obligation to do so) and the harm query we want is the same as before but measured against a different baseline—the counterfactual world where Alice gives Bob clubs. In this case the action $A = 0$ causes harm to Bob. For example, if Bob's utility is $U(y) = y$ (i.e. 1 for clubs, 0 for no clubs), then the harm caused by Alice is $P(Y_{A=1} = 1|A = 0, Y = 0) = 1$.

**Preemption Problem:** Alice robs Bob of his golf clubs. A moment later, Eve would have robbed Bob of his clubs. Therefore, Alice's action does not cause Bob to be worse off as he would have lost his clubs regardless of her actions, and so by the CCA Alice does not harm Bob by robbing him. However, intuitively Alice harms Bob by robbing him, regardless of what occurs later [43].

Let $A = \{1,0\}$ denote Alice {robbing, not robbing} Bob respectively, and similarly $E = \{1,0\}$ for Eve. $B = \{1,0\}$ denotes Bob {has clubs, does not have clubs}. Assume Bob's utility is $U(b) = b$. The causal mechanisms are $e = 1 - a$ (Eve always robs Bob if Alice doesn't) and $b = 1 - a \vee e$ (Bob has no clubs if either Alice of Eve robs him). See Figure 6 for the causal model depicting these variables.

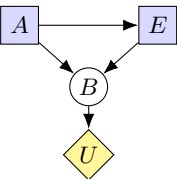

Figure 3: SCM depicting the preemption problem.

Note that while Alice's action is an actual cause of Bob not having clubs, it is also an actual cause of Eve not robbing Bob, which is an event equally as bad as Alice robbing Bob. Intuitively, when we claim that Alice robbing Bob was harmful, we are making a claims about the direct effects of Alice's actions on Bob independently of their effect on Eve's actions (independent of the effect that her action has mediated through Eves action, preventing Eve from robbing Bob), i.e we are concerned with the direct harm caused by Alice's actions on Bob (Definition 9).

The relevant harm query is the path-specific harm where we compare to the default action where Alice does not rob Bob, $\bar{a} = 0$. We want to determine the harm caused by Alice's action independently of its effect on Eve's action, which we do by blocking the path $\bar{g} = \{A \rightarrow E\}$. Applying Definition 9 amounts to replacing the mechanism for $E$ with $f^E(a) \rightarrow f_g^E(A = 1) = 0$, i.e. $E$ is evaluated for the factual value of $A$. We then compute the harm using the counterfactual default action $A = 0$, giving the counterfactual $B(A = 0, E = 0) = 1$, which gives a counterfactual utility of 1 compared to a factual utility of 0. Therefore Alice directly harmed Bob by robbing him.

Note we can also choose a different model where we explicitly represent the outcomes of the two agents decisions and the temporal order in which they occur (Figure 4). In this case the relevant harm query is essentially the same; the path specific harm where we determine the harm caused by Alice's action independently of the effect it has on whether or not Eve robs Bob (i.e. $\bar{g} = \{R_A \rightarrow R_E\}$).

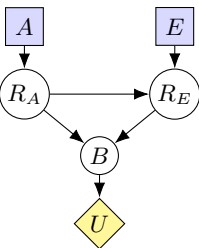

Figure 4: SCM depicting the preemption problem explicitly representing the temporal asymmetry between Alice and Eve's actions effecting Bob.

**Preventing worse:** We provide two versions of the preventing worse problem [82] which have identical causal models but intuitively different harms attributed to Alice's action.

Case 1: Bob has \$2. The thief Alice is stalking Bob in the marketplace and notices that Eve (a more effective thief) is also stalking Bob. Seeing Eve before Eve notices her, Alice decides to make her move first. She steals \$1 from Bob. Eve was going to steal \$2 from Bob, but is incapable of doing so if someone else robs him first (e.g. Bob realizes he's been robbed and call for the police, making further robbery impossible). Seeing that Bob was robbed by Alice she decides not to rob him.

Case 2: Eve has captured Bob and intends to torture him to death. Alice sees this, and is too far away to prevent Eve from doing so. She has a line of sight to Bob (but not Eve) and can shoot him before Eve has a chance to torture him to death, resulting in a painless death.

The causal model describing both of these cases is depicted in Figure 5. Let $E = \{1, 0\}$ denote if Eve is present or not, $A \in \{1, 0\}$ be Alice's action (rob, shoot) or not, $AB \in \{1, 0\}$ denote the outcome following Alice's action (Bob is robbed of \$1 / Bob is shot, or not) and let $EB \in \{1, 0\}$ denote Eves action on Bob (Bob is robbed of \$2 / Bob is tortured, or not). Let $Y \in \{0, 1, 2\}$ denote Bob's outcome, with 2 being the best (Bob has \$2 in Case 1, Bob survives in Case 2), 1 being the second worst (Bob has \$1 in Case 1, is killed painlessly in Case 2), and 0 the worst (Bob has \$0 in Case 1, died painfully in Case 2). The causal mechanisms are $a = e$ (e.g. Alice shoots/robs if Eve is present), $ab = a$ (Alice's bullet hits with certainty / successfully robs with certainty), $eb = e \wedge (1 - ab)$ (Eve tortures Bob if she is present and he is not shot / Eve robs Bob if she is present and hasn't been robbed already), and $y = ab + 2(1 - ab)(1 - eb)$ (Case 1: if Bob is shot he dies quickly, else if Eve tortures him he dies slowly, else he lives, Case 2: Bob has \$2 if not robbed, \$1 if robbed by Alice, \$0 if not robbed by Alice and robbed by Eve).

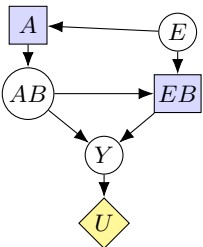

Figure 5: SCM depicting the preventing worse problem.

In Case 1, Alice intuitively harms Bob by robbing him. The argument supporting this is that Alice's robbery caused Bob to lose \$1, regardless of the fact that Alice's action prevented a worse robbery by Eve. However, for Case 2 it is argued in [82] that Alice intuitively didn't harm Bob. While Bob died due to Alice shooting him, this action was intended to prevent a worse outcome from occurring (Bob being tortured to death), which would have happened with certainty had Alice not shot him. However, these two scenarios are described by equivalent causal models—only the variables have been re-labeled.

From this we conclude that to satisfactorily describe these two situations we need two different harm queries, which we identify as the path-independent (total) and path-specific harms. This 'path dependence' of harm has been noted in psychology research, where people are more likely to

attribute harm to cases where the agent is a direct cause of that harm rather than harm occurring as a side-effect of their actions [83]. Note, this is no different than in causal analysis where in certain problems the casual effect is the desired query and in others the path-specific effect is the desired query [81]. For Case 1 we use the path-specific harm (Definition 9) to determine the harm caused by Alice robbing Bob independently of what effect it had on Eve's action. We block the path $\bar{g} = \{AB \rightarrow EB\}$ and use the default action $\bar{a} = 0$. In the counterfactual world, this gives $AB = 0$ and $EB = f^{EB}(A = 1, E = 1) = 0$, and therefore $Y = 2$, and so the direct harm of Alice robbing Bob is 2 - 1 = 1 compared to not robbing him. For Case 2, we note that while Alice shooting Bob is arguably intrinsically harmful (as is captured by the direct harm of 1 caused by $A = 1$ if we calculate the path-specific harm as in Case 1), this is not the morally relevant harm that we are referring to when we say that intuitively Alice did not harm Bob by shooting him. The reason Alice fired the shot was precisely because of its mediating effect on $Y$ through Eve's actions (preventing her from torturing him to death). From this we infer that the morally relevant harm in this case is the path-independent (total) harm. This we calculate using Definition 3 and the default action $\bar{a} = 0$, which in the counterfactual world gives $AB = 0$, $EB = 1$, $Y = 0$ and hence $U = 0$, compared to the factual utility $U = 1$, giving the desired result that Alice did not harm Bob compared to not shooting him. Note that if we favoured the path-independent or path-dependent harm a priori this would either fail to detect harm in Case 1 or incorrectly attribute harm to Alice in Case 2.

Finally, note that while we claim that the total harm is the desired harm query to evaluate the shooter example, we can also calculate the direct harm of shooting Bob (which is non-zero). This is the relevant harm query if we want to determine if shooting Bob is intrinsically a harmful action independently of what Eve will do (intuitively, shooting someone is harmful). So we can see without having to calculate anything that there is no unambiguous answer to the question 'did Alice harm Bob', it depends on if we are interested in the direct harmfulness of the action (controlling for other mediating factors) or the overall harm (including other consequences of the action, such as preventing Eve from torturing Bob to death). These are simply different types of harms, and Alice's action is either harmful or not harmful depending on which of these harms we are referring to. This is in contrast to other approaches that seek to define harm with a single causal formula that applies to all scenarios, namely [76, 77], and we discuss this approach and provide counterexamples to it in Appendix E.

## C

In this appendix we discuss some other definitions of harm besides the CCA (Definition 2).

While we have focused on the counterfactual comparative account of harm, there are alternative definitions of harm which we refer to as 'causal accounts' [84]. Broadly speaking, causal accounts define an action or event as harmful if it causes some intrinsically bad state or outcome. For example, the 'non-comparative account' by Harman [49] states "One harms someone if one causes him pain, mental or physical discomfort, disease, deformity, disability or death'. The main differences to counterfactual comparative accounts are i) causal accounts do not describe how one determines if an action caused an outcome, and ii) states or outcomes are labelled as intrinsically bad (often referred to as 'harms' [82]). The issue with i) is that counterfactuals underlie the formal methods for attributing causes to outcomes (e.g. [24]), and so simply stating 'A causes B' merely obfuscates the counterfactual comparisons necessary to make this relationship formal. The issue with ii) is that we may want to describe harms for outcomes that are not intrinsically bad. For example, if I win the lottery and someone steals half of my winnings this is arguably an act that harmed me, in spite of the fact that having half of a lottery jackpot is arguably not an intrinsically bad state to be in. To define a state as intrinsically bad, it appears that we have to choose a level of utility and claim that all states with a utility below this level are intrinsically bad and can constitute harms, and all states above this level cannot. Interestingly, this is the choice made when defining harm in a recent proposal that uses counterfactual comparisons [76] (with outcomes that obtain a utility below the default utility being bad outcomes), although the default utility her varies from case to case. Truly comparative definitions only consider the relative utility difference between the factual outcome and the counterfactual outcome (as is the case with our definition) and there is no need to introduce intrinsically bad states. We discuss this alternative definition further in appendix E.

Another alternative to counterfactual accounts of harm is to avoid causation altogether. For example the event-based account of harm [46] states that 'someone suffers a level-1 harm (benefit) with respect

to a certain basic good if and only if he loses (gains) some quantity of that good' and 'someone suffers a level-(n+1) harm with respect to a certain basic good if and only if he is prevented from receiving a level-n benefit with respect to that good'. This can be understood as harm by a loss of utility (goods), although it is incapable to attributing harm to a given cause. In our assistant example in section 4, note that this definition would make no distinction between loses caused by the assistant versus those caused by market fluctuations, or in our treatment example (section 1) would not be able to distinguish between deaths caused by doctors versus those caused by diseases. This places harm squarely at the level or gaining or losing utility and in doing so avoids providing an answer to the question of how harm should be attributed to specific actions or events (i.e. one is only capable of saying they have been harmed by something).

## D

In this Appendix we discuss selecting and interpreting default actions, harmful events, and various edge cases not covered in the main body of our paper such as harmful default actions. Note that while the CCA (Definition 2) states '[the action] had not been performed', this should not be interpreted as the counterfactual action always being 'do nothing'. Instead, we define a *harm query* as taking as input a specific default action or policy. This allows for more nuanced and general measures of harm with respect to some arbitrary baseline, much in the same way that treatment effects can in general be defined with respect to any baseline treatment (i.e. they do not always have to measure compared to 'no treatment' / a control group). While we do not provide a method for determining the desired default action or policy in general, we observe that statements about harm often implicitly assume some default action and give examples of this below. Indeed, in Appendix E we show in Example 3 that being able to enforce a unique default action is vital in some scenarios to give intuitive results, and the alternatives of assuming a single universal default action or enumerating existentially over actions can give counterintutive results. We also note that while the examples outlined in the main text assume deterministic default actions, it is trivial to extend our definitions to non-deterministic default actions by replacing $\text{do}(A = \bar{a})$ in Definition 3 with a soft intervention (e.g. [85]).

Our definition of harm treats the default action as an integral part of the harm query; harm is determined given i) an SCM of the decision task, ii) a utility function, iii) a default action, policy or event, and iv) if desired, a set of causal paths to restrict our attention to (see Appendix A). Note this is equivalent to the set of inputs when characterising treatment effects, for which we have to specify a baseline (default) treatment and, if we want to measure path-specific treatment effects, a set of permitted causal paths [21]. As with treatment effects, it falls to the practitioner to determine the desired values for these inputs to the query—the definition of the treatment effect does not inform the practitioner as to what default treatment or path-specific analysis they should choose for a specific problem. The choice to exclude specific causal pathways is usually made because we do not want to consider certain outcomes as being natural consequences of an action. For example, in Appendix B when describing the 'preventing worse' problem, if we are interested in the intrinsic harm caused by an action (e.g. the harmfulness of robbing someone, independent of its effect of the behaviour of other agents), then we restrict to the causal pathways to the utility that are not mediated by the actions of other agents.

**Note: non-comparative harm:** In our solution to the omission problem we use the fact that our definition of harm is comparative. One may be tempted to devise a non-comparative account of harm, to answer questions such as 'was $A = a$ harmful?' without in the answer having to refer to an implied baseline. We take the view that this question is ambiguous and cannot be answered in general. There are many similar questions that appear intuitive but cannot be answered without making comparisons to a baseline. For example, one can ask 'was treatment $A = a$ effective?', but effective compared to what; no treatment? Or some other alternative treatment? Often these questions appear non-comparative because they are asked within a context which implies which baseline comparison should be made. Examples of these implied comparisons are given below. While some have argued against comparative accounts on the grounds that it is not always clear which comparison is needed [46], this problem arises due to the ambiguity of statements about harm rather than due to a problem with its formal definition. Clearly, there is not a single universal comparison or default action that is suitable for all situations (this assumption leads to the omission problem, described in Appendix B), and the ability to explicitly choose the comparison is a feature rather than a fault with the CCA. While in principle our default-action-dependent measure of harm can be

converted to default-action-independent measure (as with [76, 77]) if desired, e.g. by taking the max of the harm over all default actions, in all of the examples we explore this is not desirable.

In all of the following examples we consider varations of the treatment example (Examples 1 & 2). The modelling details for these examples are given in Appendix G.

**Example 1:** *A patient is given treatment 1 and dies. Did the treatment harm the patient?* Typically, when we are asked if a given treatment is harmful we imagine a scenario where the patient is given no treatment or placebo, and ask what would have happened to the patient. If this is our interpretation of this question, then the default action is $\bar{a} = 0$ (no treatment) and the patient was not harmed, matching our intuition coming from the fact that any patients who die following treatment 1 would have died anyway.

**Example 2:** *Although standard clinical guidelines require a doctor to administer treatment 1, the doctor fails in their duty to treat the patient at all and the patient dies. Was this action harmful?* It is implied here that we want to compare to the default action $\bar{a} = 1$ (treatment 1), in which case the expected harm of the factual treatment $\bar{a} = 0$ is $> 0$.

**Example 3:** *A doctor, following standard clinical guidelines, administers treatment 2 and the patient dies due to an allergic reaction. Was the patient harmed by receiving this treatment?*. In this example, we may be tempted to choose the default treatment to be $\bar{a} = 2$ (treatment 2), as this is the normative action taken by the doctor. However, the wording suggests that we want to determine the intrinsic harm caused by administering this treatment regardless of what the doctor was expected to so, and so we should intuitively choose the default action to be $\bar{a} = 0$ which gives a non-zero harm.

**Example 4:** *A passing bystander with no duty of care to the patient decides not to treat them, and the patient dies. Did they harm the patient?* Here it is implied that the default action is not to treat the patient, and so the default action is $\bar{a} = 0$. As the factual and counterfactual actions are identical, the harm and benefit are both zero, and the bystander failed to benefit the patient rather than harm them (though this failure to benefit may also be morally reprehensible, [86]).

**Example 5:** *In a randomised control trial, participants are assigned treatment 1 or no treatment. Those who are assigned no treatment receive additional care. How beneficial is treatment 1?* There are two options here, we can calculate the total benefit i.e. 'how much benefit in total do patients given treatment 1 receive compared to those receiving the placebo', or we can calculate the direct benefit i.e. 'controlling for the fact that treated patients receive less subsequent care, how directly beneficial is treatment 1 compared to the placebo'. The question 'how beneficial is treatment 1' is ambiguous, but the most intuitive interpretation of this question is that we want to know how intrinsically beneficial the treatment is, i.e. correcting for the spurious effect of this poorly designed trial where the control group receive additional care. In this case, implied choice is that the default action is administering the placebo and we perform a path-specific analysis where we exclude the causal path from the treatment choice to additional care.

## E   Comment on Beckers et. al.

In this appendix we discuss a recent alternative proposal for defining harm using causality [76, 77]. First we describe these definitions (referred to as the BCH) and, to allow for easier comparison, describe how they can be recovered from our definitions in single action single outcome decision tasks. We then identify several examples where the BCH definitions give different results to ours and discuss why these differences arise. In doing so we identify what we believe are the following issues with the BCH definitions and give examples for each,

1. Attributing harm to actions that are not harmful (false positives)
2. Omission problem
3. Cancelling of harms v.s. benefits
4. Enforcing the worst-case harm
5. Practical limitations (intractability in complex domains / action spaces)

Finally, we address some claims about our definition and interpretation of our results reported in [76, 77].

**Definition 10** (Actual causes). *$A = a$ rather than $A = a'$ causes $Y = y$ rather than $Y = y'$ in the model $\mathcal{M}$ for exogenous noise state $E = e$ iff;*

1. *$A(e) = a$ and $Y(e) = y$*

2. *The exists a set of environment variables $W$ with factual state $W(e) = w$ such that $Y_{A=a',W=w}(e) = y'$*

3. *$A = a$ is minimal; There is no strict subset of the set of variables $\tilde{A} \subset A$ such that for $\tilde{A} = \tilde{a}$ we can satisfy conditions 1. and 2.*

In the following we will focus on scenarios where we can consider single action variables alone and so we can ignore condition 3 in Definition 10.

**Definition 11** (BCH harm). *$A = a$ harms the user in model $\mathcal{M}$ and exogenous noise state $E = e$, if there exists an outcome $Y = y$ an action $A = a'$ such that,*

H1 *$U(y) < d$ where $d$ is the default utility*

H2 *$\exists \, Y = y'$ s.t. $A = a$ rather than $A = a'$ causes $Y = y$ rather than $Y = y'$ and $U(y') > U(y)$.*

*$A = a$ strictly harms the user if in addition,*

H3 *$U(y) \leq U(y'')$ for the unique $y''$ such that $Y_{a'}(e) = y''$*

**The BCH definition in words:** Simply stated, BCH aims to define an action or event as harmful if it is an actual cause of the user having a utility lower than their default utility (we discuss 'strict harm' separately, which additionally requires H3 to be satisfied). While this is a very crisp and intuitive account of harm, we will argue that it is too permissive—concretely, that an action being an actual cause of a bad outcome is a necessary but insufficient condition for harm. As we will show there are situations where an action is an actual cause of the utility being lower than the default, which nevertheless we would not want to consider as harms.

To describe how Definition 11 is evaluated, H1 requires that the factual outcome achieved by the user is lower than some predetermined default utility (which plays a similar role to our default actions), while H2 determines if the action $A = a$ was an actual cause of this bad outcome. Following Definition 10, to determine if H2 is satisfied we consider a given exogenous noise state $E = e$ (Definition 10 determines harm for fixed noise states i.e. deterministic models), and consider every possible action $A = a'$ the agent could have taken. For each $A = a'$ we then enumerate over every subset $W$ of the endogenous variables and compute the counterfactual outcome for $A = a'$ while $W$ is held fixed in the factual state it took for $A = a$ ($W$ is referred to as a 'contingency'). This generates a counterfactual outcome for each counterfactual action and contingency, and if any of these outcomes has a higher utility than the factual outcome then $H2$ is satisfied. This method for determining harm closely mirrors the method for determining actual causes [24].

**Recovering the BCH definition from our definition:** To allow for easier comparison between the BCH definition of harm and ours, we describe how BCH can be recovered from our definition in the simplest settings by imposing additional constraints. In the case of single actions $A$ and single outcomes $Y$ (where contingencies play no role), it is possible to recover Definition 11 from ours by imposing the following additional conditions; i) restricting to deterministic models (a given $E = e$), ii) restricting to action-independent utilities $U(a, x, y) \to U(y)$, iii) calculating the counterfactual harm (using our Definition 3) for every possible default action $A = \bar{a}$ and taking the max over these values, iv) returning 'True' if the max over these values is greater than zero and the factual utility is lower than the default value $d$. In this setting the BCH definition reduces to determining if the utility is lower than some threshold and if there is non-zero harm (following our definition) for any default action. In the more general case where there are multiple actions / outcomes, our definitions diverge significantly due to the freedom in the BCH definition to choose any contingency that results in harm.

**Example 1, drowning soldier (False positives).** In this example we describe an action that is an actual cause of a bad outcome (achieving a lower utility than the default), but which we intuitively do not want to describe as harmful.

A solider falls in a river and is drowning. His officer stands beside the river and has a responsibility to help the soldier if he can. He can choose to swim out to rescue him $R = T$ or not $R = F$. There are two enemy sharpshooters watching the river. The first will shoot $S_1 = T$ if the officer tries to rescue him and wont shoot otherwise, while the second will shoot $S_2 = T$ if he doesn't try to rescue him, and wont shoot if he does. The officer decides not to rescue the soldier and he is shot dead $D = T$ by shooter 2. Was the officers decision not to rescue the soldier harmful?

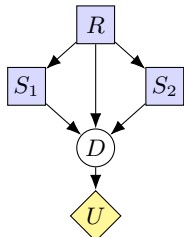

Figure 6: Shooter problem

The causal equations for this situation in Boolean notation are $S_1 = R$ (shooter 1 shoots if rescue attempt), $S_2 = \neg R$ (shooter 2 shoots iff no rescue attempt, $\neg =$ 'NOT'), $D = \neg[R \wedge \neg S_1 \wedge \neg S_2]$ (the soldier doesnt die if he is rescued and shooter 1 and 2 do not shoot, otherwise he dies). Let $U(D = T) = 0$ and $U(D = F) = 1$. Clearly, shooter 2 harmed the soldier, so we require the default utility to be $d > 0$ otherwise BCH would assign no harm to shooting the soldier. However, the officers choice not to rescue the soldier is also an actual cause of the soldier dying. To see this, note that if we take the contingency where we hold $S_1 = F$ fixed in its factual state, then the counterfactual outcomes for $R = T$ are $S_2 = F$ (the second shooter wont shoot) and (as $S_1 = F$) then $D = F$ (the soldier survives). As the default utility is greater than 0, we conclude that the officer harmed the soldier by not attempting the rescue him, despite the fact that this outcome never could have occurred (in reality, the soldier would have died regardless of what the officer did).

Intuitively, shooter 1 harmed the soldier by killing them, and the officer failed to benefit the soldier by failing to prevent this. This is what we find using our definition of the total harm. However, both the officer's decision and the action of shooter 1 are actual causes of the soldier dying. This highlights that being an actual cause of a bad outcome is a necessary but insufficient condition for harming. By allowing for arbitrary contingencies (as is required when attributing actual causes) we end up attributing harm to the officer who intuitively did no harm (rather, they failed to benefit). This is somewhat analogous to computing the path-specific harm over all subsets of paths and taking the max over these harms.

**Example 2, Omission problem (False positives).** The omission problem is another example of a false positive where an agent is mistakenly attributed harm because they failed to take a beneficial action. In the standard omission problem (see Appendix B) there is a single agent, and BCH gives the correct answer in this case because we can choose the default utility to be zero. In this example we present an extension of the omission problem that is violated by BCH.

Alice can choose to give Bob her golf clubs or not. She has no obligation to do so. Unbeknownst to her, Eve is planning to rob Bob, but if Bob is holding a golf club she will not dare rob him. Alice decides not to give Bob her golf club and Bob is robbed by Eve. By choosing not to gift her clubs, did Alice harm Bob?

There are four possible outcomes; Bob has / does not have clubs $(C)$, Bob is / is not robbed $(R)$. Let $R \times C$ be the packaged-up outcome variable, with $U(R = T, C = F) < U(R = F, C = F) < U(R = F, C = T)$. Clearly, we require Eve's action to be harmful so the default utility must be $d > U(R = T, C = F)$. However, if Alice had given Bob the clubs then we would have the counterfactual outcome $R = F, C = T$. Therefore the BCH definition asserts that Alice harmed Bob by not giving him her clubs. However, another more intuitive perspective is that Alice failed to benefit Bob, rather than harm him (just as in the omission problem). The issue with the BCH here is twofold; by using default utilities and having to accomodate that Eve harmed Alice, we cannot use the BCH solution for the standard omission problem where we choose the default utility to always correspond to Bob having 'no clubs'. Then, as we must consider all possible counterfactual actions,

we must conclude that Alice harmed Bob. With our definition this problem never arises, we simple choose the obvious default action to be that Alice does not give Bob her clubs.

**Example 3, Cancelling harms and benefits.** While in [76] the default utility is often chosen to reflect some actual utility the user can receive, this is not necessarily the case, and to get the BCH definition to give the desired answer we sometimes have to choose a default utility that is hard interpret or choose ahead of time (beyond selecting whatever default utility gives the desired solution). Consider the following example where any choice of default utility gives a counterintutive result.

Bob has \$1. Alice hacks into Bobs credit card and robs \$$K$ from him. Seeing this, Bobs colleague Eve takes pity on him and gives him \$$K$. We claim that Alice's action was directly harmful and Eves action was beneficial. An intuitive property of harm is that the harms caused by one agent's actions should not by default be cancelled out by the another agents beneficial actions—stealing from someone is harmful, regardless of whether or not someone else responds by reimbursing the victim. Even if we are in a state of high utility, typically there are many events in the past that contributed to this outcome that were harmful or beneficial, and it is important to be able to identify and quantify these harms and benefits with respect to our current utility (for example we may want to punish Alice in proportion to how harmful her action was, and reward Eve similarly). By analogy, the total causal effect of Alice's action on Bob's utility is zero, but the direct causal effect is non-zero.

Using our path-specific harm (Definition 9) and a default action of $K = 0$ for Alice, we measure the direct harm of Alice robbing \$$K$ from Bob to be $K$. Can we do this with the BCH definition? In this example, if we make the obvious choice $d = 1$ (Bob expects to have a dollar), then H1 is not satisfied and Alice's theft was not harmful. In the BCH definition, no actions or events are harmful if the user's utility is greater or equal to the default. Therefore we must choose $d > 1$, although this is difficult to justify or interpret (should Bob expect to have more money than he has / he started with?). Using Definition 12 we compare to the highest utility counterfactual, which is the outcome where where we fix the contingency that Eve gives Bob \$$K$ and choose the counterfactual action where Alice doesn't rob Bob. This gives a counterfactual utility of $1 + K$, and so the harm according to Definition 12 is $\max(0, \min(d, 1 + K) - 1)$. Therefore to recover that stealing \$$K$ has a harm of $K$, we must choose $d \geq 1 + K$. This default utility can be made arbitrarily large by choosing larger values of $K$, and in general $d$ should be defined independently of Alice's action (i.e. her choice of $K$). One solution to this problem would be to use the intuitive default utility $d = 1$ along with a path-specific version of the BCH definition of harm, although it is not immediately clear how to formalise this path-specific version as this would require a path-specific variant of Definition 10 which in its current form considers all contingencies.

**Example 4, enforcing worst-case harm.** In [77] the authors present a quantitative definition of harm,

**Definition 12** (BCH quantitative harm). *If for a given $E = e$, $A = a$ rather than $A = a'$ causes $Y = y$ rather than $Y = y'$, then the quantiative harm of $A = a, Y = y$ relative to $A = a', Y = y'$ is,*

$$QH(a, y, a', y', e, ; \mathcal{M}) = \max(0, \min(d, U(y') - U(y)))$$

*where $d$ is the default utility. The quantitative harm for $E = e$ caused by $A = a$ is*

$$QH(a, e, ; \mathcal{M}) = \max_{a', e'} QH(a, y, a', y', e, ; \mathcal{M})$$

*if there is some $A = a'$ and $Y = y'$ such that $A = a$ rather than $A = a'$ causes $Y = y$ rather than $Y = y'$. If there is no such $A = a', Y = y'$ then the quantitative harm caused by $A = a$ is zero.*

We refer to Definition 12 as the *max harm*, as for each $E = e$ it compares to the counterfactual action and contingency that gives the highest counterfactual utility. In [77] the expected harm (averaging over $E = e$) is sometimes computed using a weighting scheme, although it is not clear how these weights are chosen in a systematic way. However if there is some $P(E = e) > 0$ for which $QH(a, e, ; \mathcal{M}) > 0$ then we will claim that the harm is greater than zero by Definition 12 .

As noted in [77] the max harm is non-zero for treatment 1 in our introductory example (see Examples 1 & 2 in the main text, and Appendix G). This is in spite of the fact that for all patients (for every $E = e$) treatment 1 increases the survival rate compared to no treatment. The max harm for treatment 1 is greater than zero because there are some patients who die given treatment 1 who would survive if they had been given treatment 2. Essentially, we are forced to measure harm compared to the best possible action we could have taken in every noise state.

While the max harm is likely the desired measure of harm in some situations (for example, a doctor who can observe $E = e$ may be expected to always take the best possible action for a patient), it does not permit the standard perspective used for measuring the harms and benefits of medicines (which is the situation described in our example). The intrinsic harms of medicines are often described in terms of risks, where the rates of a negative outcome following treatment are compared the rates for a control / placebo group (see for example the 'number needed to harm (NNT)' in [87]), not by comparing to the rates for a group where every patient is given the optimal treatment for that individual. Note that to get from harms to risks, one has to make the assumption of counterfactual independence[2].

Medicines typically have heterogeneous effects, and if there is some sub-population $E = e$ where treatment 1 is not as effective as treatment 2, it is typically claimed that treatment 1 is less effective (less beneficial) for that sub-population that treatment 2, rather than claiming that treatment 1 is harmful for that population (that is, unless it is less effective than 'no treatment'). In our treatment example, this is captured by measuring harm as 'the number of patients who die who would have lived if they had not been treated'. This is not the only way of measuring harm for treatments (we discuss several others in Appendix D), but it is the standard perspective when describing the harms and benefits of medicines. By forcing us to consider every possible action and contingency, the max-harm is incapable of accommodating this perspective.

**Practical limitations:** In addition to the limitations of our definition (namely knowledge of the SCM, which we discuss in Appendix L.1), we comment on some additional limitations specific to the BCH definition of harm. Firstly (as noted by the authors [76, 77]) calculating the harm by the BCH definition is intractable. This is because to evaluate Definition 10 we must enumerate existentially over possible actions and contingencies (evaluating Definition 11 is at least DP-complete [88]). As the authors point out in [76], this practically restricts the BCH definition to situations with a few low-dimensional variables. The authors argue that most causal models we will use to deal with harm in the real world will only consist of only a few low-dimensional variables. This does not appear to be the case at first glance, for example, even the simple dose response model we use to illustrate our results in section 6 has a continuous action and outcome space, and so does not fit the description of models compatible with BCH. This model is very minimal compared to a real-world clinical decision task which typically involve a large number of outcome variables that can be continuously valued and/or include many (potentially sequential) actions. Ultimately, restricting to decision tasks that have an efficiently enumerable action space and outcome space prevents the BCH definition from being applied to most machine learning settings, where this condition is typically not satisfied (indeed one of the main motivations for using machine learning algorithms in these domains is that we cannot efficiently enumerate over all possible combinations of states and actions).

**Discussion of our definition of harm in Becker's et. al.** The main criticisms of our work in [76, 77] are as follows; i) Our definition makes use of but-for causation rather than Definition 10, ii) our definition assumes the default action is determined normatively and would assign no harm to the scenario where a patient is given a standard (normative) treatment which unintentionally harms them, iii) that we introduce the possibility for a path-specific analysis to 'deal with the limitations of but-for causation' and do not explain how to choose when a path-specific analysis is necessary to determine harm. We address these points below.

i) While but-for counterfactuals alone are insufficient for attributing actual causes, we argue that an action or event being an actual cause of a bad outcome is a necessary but insufficient condition for harm, and have shown that treating harm attribution as actual cause attribution actually leads to the wrong answers. We point out that but-for counterfactuals are also widely used for causal reasoning in other areas, for example in causal inference [21] and causal definitions of fairness [33].

ii) The authors claim that our definition assigns zero harm to any action that is 'morally preferable'. This is clearly not a claim that we make as can be seen by our introductory treatment example, where the default action is 'no treatment' which is clearly not the morally

---

[2]To see how harm can be treated as risk in clinical trials, note also that the harm reduces to the risk in this example if we assume counterfactual independence, $P(Y_{a^*}^* | Y_a) = P(Y_{a^*}^*)$. This can be understood as the standard assumption in clinical trials that (for example), if a patient is given the treatment and dies, then the probability they would have lived if they had not been treated is simply the probability of death in the untreated group.

preferable action. Consider this example but where treatment 1 is no longer available. Treatment 2 may be the morally preferable action, as it achieves a higher recovery rate than no treatment, and our definition assigns non-zero harm to treatment 2. The broader point made by the authors is that for our definition the default action is not harmful. This is a misconception; we do not claim that the default action is not harmful, we claim that any action *is not harmful compared to its self*, which is trivially true. The default action is merely a comparative baseline, not a normatively determined 'morally preferable' action, as explained in Appendix D. By analogy, for a treatment $T$ the average treatment effect is zero when setting the baseline treatment to be $T$ also. Secondly, we point out that in their definition any action that achieves the default utility is not harmful, which unlike the default action having zero harm is a non-trivial assumption which results in bona fide counterexamples to their definition as outlined in this section.

iii) We do not use path-specific analysis to deal with the limitations of but-for causation (see Appendix D for discussion). Additionally, our use of path-specific analysis is directly comparable to its well-established use in path-specific effects [81]. Just as with our definition of harm, these path-specific effects include i) path-specific variants and ii) variable base-line (i.e. default) treatments, and at no point in their definition is the baseline treatment or the causal path specified—these are inputs to the query (as is the default utility in the BCH definition).

## F

In this Appendix we prove Theorem 1. Noting that $\max\{0, U(a, x, y) - U(\bar{a}, x, y)\} - \max\{0, U(\bar{a}, x, y^*) - U(a, x, y)\} = U(a, x, y) - U(\bar{a}, x, y^*)$, subtracting the expected harm from the expected benefit (Def 3) gives,

$$\mathbb{E}[b|a, x; \mathcal{M}] - \mathbb{E}[h|a, x; \mathcal{M}] \tag{15}$$

$$= \int_{y, y^*} P(y, Y_{\bar{a}} = y^*|a, x; \mathcal{M})(U(a, x, y) - U(\bar{a}, x, y^*)) \tag{16}$$

$$= \int_y P(y|a, x)U(a, x, y) - \int_{y^*} P(Y_{\bar{a}} = y^*|x)U(\bar{a}, x, y^*) \tag{17}$$

$$= \mathbb{E}[U|a, x] - \mathbb{E}[U|\bar{a}, x] \tag{18}$$

## G

In this Appendix we derive the SCM model for the treatment decision task in examples 1 and 2, and calculate the average treatment effect and counterfactual harm.

Patients who receive the default 'no treatment' $T = 0$ have a 50% survival rate. $T = 1$ has a 60% chance of curing a patient, and a 40% chance of having no effect, with the disease progressing as if $T = 0$, whereas $T = 2$ has a 80% chance of curing a patient as a 20% chance of killing them, due to some unforeseeable allergic reaction to the treatment.

Next we evaluate this expression for our two treatment by constructing an SCM for the decision task. The patient's response to treatment is described by three independent latent factors (for example genetic factors) that we model as exogenous variables. Firstly, half of the patients exhibit a robustness to the disease which means they will recover if not treated, which we encode as $E^1 \in \{0, 1\}$ where $e^1 = 1$ implies robustness with $P(e^1 = 1) = 0.5$. Secondly, the patients may exhibit a resistance to treatment 1 indicated by variable $E^2$, with $e^2 = 1$ implying resistance with $P(e^2 = 1) = 0.4$. Finally, the patients can be allergic to treatment 2, indicated by variable $E^3$ with $e^3 = 1$ and $P(e^3 = 1) = 0.2$. Given knowledge of these three factors the response of any patient is fully determined, and so we define the exogenous noise variable as $E^Y = E^1 \times E^2 \times E^3$ with $P(e^Y) = P(e^1)P(e^2)P(e^3)$.

Next we characterise the mechanism $y = f(t, e^Y) = f(t, e^1, e^2, e^3)$ where $f(0, e^Y) = [e^1 = 1]$ (untreated patients recover if they are robust), $f(1, e_Y) = [e^1 = 1] \vee [e^2 = 0]$ (patients with $T = 1$ recover if they are robust or non-resistant) and $f(2, e_Y) = [e^3 = 0]$ (patients with $T = 2$ recover if

they are non-allergic), where $[X = x]$ are Iverson brackets which return 1 if $X = x$ and 0 otherwise, and $\vee$ is the Boolean OR.

The recovery rate for $T = 1$ and $T = 2$ can be calculated with (1) to give $P(Y_1 = 1) = P(e^1 = 1 \vee e^2 = 0) = 1 - P(e^1 = 0)P(e^2 = 1) = 0.8$, and likewise $P(Y_2 = 1) = P(e^3 = 0) = 0.8$. Hence the two treatments have identical outcome statistics (recovery/mortality rates), and all observational and interventional statistical measures are identical, such as risk, expected utility and the effect of treatment on the treated. Note as there are no unobserved confounders the recovery rate for action $A = a$ is equal to $\mathbb{E}[Y_a]$.

We compute the counterfactual expected harm by evaluating (4), noting that $Y_0^*(e) = 1$ if $e^1 = 1$, $Y_1^*(e) = 0$ if $e^1 = 0$ and $e^2 = 1$, and $Y_2^*(e) = 0$ if $e^3 = 1$. This gives $P(Y_1 = 0, Y_0^* = 1) = 0$, i.e. there are no values of $e^Y$ that satisfy both $Y_1(e) = 0$ and $Y_0(e) = 1$, and therefore $\mathrm{do}(T_1 = 1)$ causes zero harm. However, $P(Y_2 = 0, Y_0^* = 1) = P(e^1 = 1)P(e^3 = 1) = 0.1$, and so $\mathrm{do}(T_2 = 2)$ causes non-zero harm. This is due to the existence of allergic patients who are also robust, and will die if treated with $T = 2$ but would have lived had $T = 0$.

# H

In this Appendix we derive the policies of agents 1-3 in Example 3. We note that outcome $Y$ is described by a heteroskedastic additive noise model with the default action $\bar{a}$ (no action) corresponding to $A = 1, K = 1$. The expected harm is given by Theorem 5 with $\sigma(\bar{a}) = 100$, $\sigma(A = 2) = 100$, $\sigma(A = 3) = 0$ and $\sigma(A = 1, K) = 100K$. $\mathbb{E}[U|\bar{a}] = 100$ $\mathbb{E}[U|A = 2] = 110$, $\mathbb{E}[U|A = 3] = 80$ and $\mathbb{E}[U|A = 1] = 100K$, where we have used $\mathrm{Var}(KY) = K^2\mathrm{Var}(Y)$ and $\mathrm{Var}(Y + 10) = \mathrm{Var}(Y)$

Agent 1 takes action 1 and the maximum value $K = 20$ as this extremizes $\mathbb{E}[U|a]$.

Agent 2 chooses $a = \arg\max_a\{\mathbb{E}[Y|a] - \mathrm{Var}(Y|a)\}$ which for each action is given by,

$$E[Y|A = 1] - \lambda\mathrm{Var}(Y|A = 1) = 100K - 100^2 K^2 \lambda \tag{19}$$

$$E[Y|A = 2] - \lambda\mathrm{Var}(Y|A = 2) = 110 - 100^2\lambda \tag{20}$$

$$E[Y|A = 3] - \lambda\mathrm{Var}(Y|A = 3) = 80 \tag{21}$$

For action 1 the optimal $K = 1/200\lambda$, which gives $E[Y|A = 1] - \mathrm{Var}(Y|A = 1) = 1/4\lambda$. Note that $1/4\lambda > 110 - 100^2\lambda$ for $\lambda < 0.0032$, which $80 > 1/4\lambda$ for $\lambda > 0.003125$. Therefore there is no value of $\lambda$ for which agent 2 selects action 2, choosing action 1 for $\lambda < 0.003125$ and action 3 otherwise.

For agent 3 applying Theorem 5 gives,

$$\mathbb{E}[Y|A = 1] - \lambda\mathbb{E}[h|A = 1, K] = 100K - \lambda\left[\frac{|100(K-1)|}{\sqrt{2\pi}}e^{-\frac{1}{2}} + \frac{100(K-1)}{2}\left(\mathrm{erf}\left(\frac{\mathrm{sign}(K-1)}{\sqrt{2}}\right) - 1\right)\right] \tag{22}$$

$$= \begin{cases} 100K - 8.332(K-1)\lambda, & K \geq 1 \\ 100K - 59.937(1-K)\lambda, & K < 1 \end{cases} \tag{23}$$

$$E[Y|A = 2] - \lambda\mathbb{E}[h|A = 2] = 110 \tag{24}$$

$$E[Y|A = 3] - \lambda\mathbb{E}[h|A = 3] = 80 - \lambda\left[\frac{100}{\sqrt{2\pi}}e^{-\frac{20^2}{2\times100^2}} + \frac{20}{2}\left(\mathrm{erf}\left(\frac{20}{\sqrt{2}\times100}\right) - 1\right)\right] \tag{25}$$

Clearly, the agent will never take action 3 as its expected HPU is smaller than that for action 2 for all $\lambda$. For action 1, for $K < 1$ the expected HPU is also smaller that that for action 2, for all $\lambda$. For action 1 with $K > 1$, if $\lambda < 12.002$ the optimal $K = 20$, otherwise it is 0. As a result, for $\lambda < 11.93$ the agent chooses action 1 with $K = 20$, and otherwise chooses action 2.

# I

In this Appendix we derive an expression for the expected counterfactual harm in generalized additive models. To calculate the expected counterfactual harm we derive a solution for a broad class of SCMs, heteroskedastic additive noise models, which includes our GAM (11),

**Definition 13** (Heteroskedastic additive noise models). *For $Y$, $Pa(Y) = A \cup X$, the mechanism $y = f_Y(a, x)$ is a heteroskedastic additive noise model if $Y$ is normally distributed with a mean and variance that are functions of $a, x$,*

$$y = \mu(a, x) + e^Y \sigma(a, x), \quad e^Y \sim \mathcal{N}(0, 1) \tag{26}$$

In Appendix J we show that the dose response model (11) can be parameterised as a heteroskedastic additive noise model and calculate the expected counterfactual harm using the following theorem,

**Theorem 5** (Expected harm for heteroskedastic additive noise model). *For $Y = f_Y(a, x, e^Y)$ where $f_Y$ is a heteroskedastic additive noise model (Definition 13) and default action $A = \bar{a}$, the expected harm is*

$$\mathbb{E}[h|a, x] = \frac{|\Delta\sigma|}{\sqrt{2\pi}} e^{-\frac{\Delta U^2}{2\Delta\sigma^2}} + \frac{\Delta U}{2} \left( \text{erf}\left( \frac{\Delta U}{\sqrt{2}|\Delta\sigma|} \right) - 1 \right) \tag{27}$$

*where* $\text{erf}(\cdot)$ *is the error function,* $\Delta U = \mathbb{E}[U|a, x] - \mathbb{E}[U|\bar{a}, x]$, $\Delta\sigma = \sigma(a, x) - \sigma(\bar{a}, x)$.

*Proof.* Note that if $e^Y \sim \mathcal{N}(\mu, V)$ we can replace $e^Y \rightarrow e'^Y = e^Y/\sqrt{V} - \mu$ and absorb these terms into $f(a, x)$ and $\sigma(a, x)$. Hence we need only consider zero-mean univariate noise. In the following we use $e^Y = \varepsilon \sim \mathcal{N}(0, 1)$ to denote the fact the the exogenous noise term is univariate normally distributed. We also use the fact that there are no unobserved confounders between $A$ and $Y$ to give $P(y|a, x) = P(y_a|x)$. Calculating the expected counterfactual harm using gives

$$\mathbb{E}[h|a, x] = \int_y dy \int_{y^*} dy^* P(y, Y_{\bar{a}} = y^*, |a, x) \max(0, U(\bar{a}, x, y^*) - U(a, x, y)) \tag{28}$$

$$= \int_y dy \int_{y^*} dy^* P(Y_a = y, Y_{\bar{a}} = y^*|x) \max(0, U(\bar{a}, x, y^*) - U(a, x, y)) \tag{29}$$

$$= \int_\varepsilon P(\varepsilon) d\varepsilon \int_y dy \int_{y^*} dy^* P(Y_a = y, Y_{\bar{a}} = y^*|\varepsilon, a, x) \max(0, U(\bar{a}, x, y^*) - U(a, x, y)) \tag{30}$$

$$= \int_\varepsilon P(\varepsilon) d\varepsilon \int_y dy \int_{y^*} dy^* P(Y_a = y|\varepsilon, a, x) P(Y_{\bar{a}} = y^*, |\varepsilon, a, x) \max(0, U(\bar{a}, x, y^*) - U(a, x, y)) \tag{31}$$

Substituting in $U(a, x, y) = y$ and $P(y|\varepsilon, a, x) = \delta(y - f(a, x) - \varepsilon\sigma(a, x))$ gives,

$$\mathbb{E}[h|a, x] = \int d\varepsilon P(\varepsilon) \max\{0, f(\bar{a}, x) - f(a, x) + \varepsilon(\sigma(\bar{a}, x) - \sigma(a, x))\} \tag{32}$$

$$= \int d\varepsilon P(\varepsilon) \max(0, -(\mathbb{E}[U|a, x] - \mathbb{E}[U|\bar{a}, x]) - \varepsilon(\sigma(a, x) - \sigma(\bar{a}, x))) \tag{33}$$

where we have used the fact that $\mathbb{E}[U|a, x] = \int d\varepsilon P(\varepsilon)(f(a, x) + \varepsilon\sigma(a, x)) = f(a, x)$. For ease of notation we use $\Delta U = E[U|a, x] - E[U|\bar{a}, x]$, $\Delta\sigma = \sigma(a, x) - \sigma(\bar{a}, x)$. Next, we remove the max() by incorporating it into the bounds for the integral. If $\Delta U > 0$ and $\Delta\sigma > 0$, this is equivalent to $\varepsilon < -\Delta U/\Delta\sigma$ and hence,

$$\mathbb{E}[h|a,x] = \int\limits_{\varepsilon < -\Delta U/\Delta\sigma} d\varepsilon P(\varepsilon)\left(-\Delta U - \varepsilon\Delta\sigma\right) \tag{34}$$

$$= -\Delta U \int\limits_{-\infty}^{-\Delta U/\Delta\sigma} P(\varepsilon)d\varepsilon - \Delta\sigma \int\limits_{-\infty}^{-\Delta U/\Delta\sigma} \varepsilon P(\varepsilon)d\varepsilon \tag{35}$$

$$\tag{36}$$

Using the standard Gaussian integrals

$$\int_a^b P(\varepsilon)d\varepsilon = \frac{1}{2}\left[\mathrm{erf}(\frac{b}{\sqrt{2}}) - \mathrm{erf}(\frac{a}{\sqrt{2}})\right] \tag{37}$$

$$\int_a^b \varepsilon P(\varepsilon)d\varepsilon = P(a) - P(b) \tag{38}$$

where $P(\varepsilon) = e^{-\varepsilon^2/2}/\sqrt{2\pi}$ and $\mathrm{erf}(z)$ is the error function, we recover

$$\mathbb{E}[h|a,x] = \frac{-\Delta U}{2}\left[\mathrm{erf}(\frac{-\Delta U}{\sqrt{2}\Delta\sigma}) - \mathrm{erf}(-\infty)\right] - \Delta\sigma\left[P(-\infty) - P(-\Delta U/\Delta\sigma)\right] \tag{39}$$

$$= \frac{\Delta U}{2}\left[\mathrm{erf}\left(\frac{\Delta U}{\sqrt{2}\Delta\sigma}\right) - 1\right] + \frac{\Delta\sigma}{\sqrt{2\pi}}e^{-\frac{\Delta U^2}{2\Delta\sigma^2}} \tag{40}$$

where we have used $\mathrm{erf}(-z) = -\mathrm{erf}(z)$ and $P(-z) = P(z)$. Similarly, if $\Delta U > 0$, $\Delta\sigma < 0$ then the max() in (33) can be replaced with a definite intergral over $\varepsilon > \Delta U/\Delta\sigma$ giving,

$$\mathbb{E}[h|a,x] = \int\limits_{\varepsilon > \Delta U/\Delta\sigma} d\varepsilon P(\varepsilon)\left(-\Delta U - \varepsilon\Delta\sigma\right) \tag{41}$$

$$= -\Delta U \int\limits_{-\Delta U/\Delta\sigma}^{\infty} P(\varepsilon)d\varepsilon - \Delta\sigma \int\limits_{-\Delta U/\Delta\sigma}^{\infty} \varepsilon P(\varepsilon)d\varepsilon \tag{42}$$

$$= -\frac{\Delta U}{2}\left[\mathrm{erf}(\infty) - \mathrm{erf}(\frac{-\Delta U}{\sqrt{2}\Delta\sigma})\right] - \Delta\sigma\left[P(\frac{-\Delta U}{\sqrt{2}\Delta\sigma}) - P(\infty)\right] \tag{43}$$

$$= \frac{\Delta U}{2}\left[\mathrm{erf}\left(\frac{\Delta U}{\sqrt{2}|\Delta\sigma|}\right) - 1\right] + \frac{|\Delta\sigma|}{\sqrt{2\pi}}e^{-\frac{\Delta U^2}{2\Delta\sigma^2}} \tag{44}$$

Next, if $\Delta U < 0$ and $\Delta\sigma > 0$ we recover the same integral as (35), and if $\Delta U < 0$ and $\Delta\sigma < 0$ we recover the same integral as (41). Hence the general solution for all $\Delta\sigma$ is (44).

$\square$

# J

In this Appendix we present the GAM dose response model including parameter values, and show that it corresponds to a heteroskedastic additive noise model and calculate the expected harm for a given dose.

We follow the set-up described in [69], where outcome $Y$ denotes the level of improvement in the symptoms of schizoaffective patients following treatment and compared to pre-treatment levels, measured in terms of the Positive and Negative Syndrome Scale (PANSS) [89]. The response of $Y$ w.r.t dose $A$ (Aripiprazole mg/day) is determined using a generalized additive model fit with a cubic splines regression and random effects,

$$y = \theta_1 a + \theta_2 f(a) + \varepsilon_0 \tag{45}$$

where the parameters $\theta_i$ are random variables $\theta_i \sim \mathcal{N}(\hat{\theta}_i, V_i)$, $\varepsilon_0 \sim \mathcal{N}(0, V_0)$ is the sample noise, and the spline function $f(a)$ is given by,

$$f(a) = \frac{(a - k_1)_+^3 - \frac{k_3 - k_1}{k_3 - k_2}(a - k_2)_+^3 + \frac{k_2 - k_1}{k_3 - k_2}(a - k_3)_+^3}{(k_3 - k_1)^2} \tag{46}$$

where $k_1, k_2, k_3$ are the knots at $a = 0, 10$ and $30$ respectively, with $(u)_+ = \max\{0, u\}$. In the following we assume for simplicity that $\theta_1$ and $\theta_2$ are independent. This hierarchical model can be expressed as an SCM with the mechanism for $Y$ given by,

$$y = \left( \hat{\theta}_1 a + \hat{\theta}_2 f(a) \right) + \varepsilon_1 a + \varepsilon_2 f(a) + \varepsilon_0 \tag{47}$$

where $\varepsilon_i \sim N(0, V_i)$. We will now reparameterise this as an equivalent SCM that is an additive heteroskedastic noise model. Using the identifies $Z = kY$, $Y \sim \mathcal{N}(0, 1) \implies Z \sim \mathcal{N}(0, k^2)$, and $Z = X + Y$, $X \sim \mathcal{N}(0, V_X)$, $Y \sim \mathcal{N}(0, V_Y) \implies Z \sim \mathcal{N}(0, V_X + V_Y)$ (where $V_X$ is the variance of $X$ and likewise for $V_Y, Y$), we can replace $\varepsilon_1 a + \varepsilon_2 f(a) \to \varepsilon g(a)$ where $\varepsilon \sim \mathcal{N}(0, 1)$ and $g(a) = \sqrt{a^2 V_1 + f(a)^2 V_2}$. We can therefore reparameterise the mechanism for $Y$ as

$$y = \mathbb{E}[U|a] + g(a)\varepsilon + \varepsilon_0 \tag{48}$$

where we have used $U(a, x, y) = U(a, y) = y$ and the fact that $\varepsilon$, $\varepsilon_0$ are mean zero to give $\mathbb{E}[U|a] = \theta_1 a + \theta_2 f(a)$. Finally, we note that the sample noise term $\varepsilon_0$ cancels in the expression for the harm,

$$\mathbb{E}[h|a] = \int_y dy \int_{y^*} dy^* P(y, Y_{\bar{a}} = y^*, |a) \max\left(0, U(\bar{a}, y^*) - U(a, y)\right) \tag{49}$$

$$= \int_y dy \int_{y^*} dy^* P(Y_a = y, Y_{\bar{a}} = y^*) \max\left(0, U(\bar{a}, y^*) - U(a, y)\right) \tag{50}$$

$$= \int_\varepsilon P(\varepsilon)d\varepsilon \int_{\varepsilon_0} P(\varepsilon_0)d\varepsilon_0 \int_y dy \int_y dy^* P(Y_a = y, Y_{\bar{a}} = y^*|\varepsilon, \varepsilon_0, a) \max\left(0, U(\bar{a}, y_{\bar{a}}^*) - U(a, y)\right) \tag{51}$$

$$= \int_\varepsilon P(\varepsilon)d\varepsilon \int_{\varepsilon_0} P(\varepsilon_0)d\varepsilon_0 \int_y dy \int_{y^*} dy^* P(y, |\varepsilon, \varepsilon_0, a) P(Y_{\bar{a}} = y^*, |\varepsilon, \varepsilon_0) \max\left(0, U(\bar{a}, y^*) - U(a, y)\right) \tag{52}$$

Substituting in $P(Y_a = y | \varepsilon, \varepsilon_0) = \delta(y - f(a) + g(a)\varepsilon + \varepsilon_0)$ gives,

$$\mathbb{E}[h|a] = \int_\varepsilon P(\varepsilon)d\varepsilon \int_{\varepsilon_0} P(\varepsilon_0)d\varepsilon_0 \max\left(0, f(\bar{a}) + g(\bar{a})\varepsilon + \varepsilon_0 - f(a) - g(a)\varepsilon - \varepsilon_0\right) \tag{53}$$

$$= \int_\varepsilon P(\varepsilon)d\varepsilon \int_{\varepsilon_0} P(\varepsilon_0)d\varepsilon_0 \max\left(0, f(\bar{a}) - f(a) + (g(\bar{a}) - g(a))\varepsilon\right) \tag{54}$$

$$= \int_\varepsilon P(\varepsilon)d\varepsilon \max\left(0, f(\bar{a}) - f(a) + (g(\bar{a}) - g(a))\varepsilon\right) \tag{55}$$

Therefore we can ignore the sample noise term when calculating the expected harm, instead calculating the expected harm for the model $Y = f(a) + g(a)\varepsilon$. This is a heteroskedastic additive noise model, and therefore by Theorem 5 the expected harm is,

$$\mathbb{E}[h|a] = \frac{\Delta U}{2}\left[\text{erf}\left(\frac{\Delta U}{\sqrt{2}\Delta\sigma}\right) - 1\right] + \frac{\Delta\sigma}{\sqrt{2\pi}}e^{-\Delta U^2/2\Delta\sigma^2} \tag{56}$$

where $\Delta U = \mathbb{E}[U|a] - \mathbb{E}[U|\bar{a}]$, $\Delta\sigma = g(a) - g(\bar{a})$ and $g(a) = \sqrt{a^2 V_1 + f(a)^2 V_2}$

The resulting curves prefented in Figure 2 are calculated using (56) and the parameter values taken from [69] (Table 1), which are fitted in a meta-analysis of the dose-responses reported in [90, 91, 92, 93, 94].

Table 1: Parameters for the hierarchical generalized additive dose-response model reported in [69]

| Parameter | Value |
|---|---|
| $\hat{\theta}_1$ | 0.937 |
| $\hat{\theta}_2$ | $-1.156$ |
| $V_1$ | 0.03 |
| $V_2$ | 0.10 |

# K

In this Appendix we present proofs of Theorems 2, 3 and 4. First, we prove Theorem 2.

**Theorem 2:** For any utility functions $U$, environment $\mathcal{M}$ and default action $A = \bar{a}$ the expected HPU is never a harmful objective for $\lambda > 0$.

*Proof.* Let $a_{\max} = \arg\max_a\{\mathbb{E}[U|a, x] - \lambda\mathbb{E}[h|a, x; \mathcal{M}]\}$. If $\exists\, a' \neq a_{\max}$ such that $\mathbb{E}[U|a', x] \geq \mathbb{E}[U|a_{\max}, x]$ and $\mathbb{E}[h|a', x; \mathcal{M}] < \mathbb{E}[h|a_{\max}, x; \mathcal{M}]$, then $\mathbb{E}[U|a_{\max}, x] + \lambda\mathbb{E}[h|a_{\max}, x; \mathcal{M}] < \mathbb{E}[U|a', x] + \lambda\mathbb{E}[h|a', x; \mathcal{M}]\,\forall\,\lambda > 0$ and so $a_{\max} \neq \arg\max_a\{\mathbb{E}[U|a, x] - \lambda\mathbb{E}[h|a, x; \mathcal{M}]\}$. $\square$

Next, we prove theorems 3 and 4 by example, constructing distributional shifts that reveal if an objective function is harmful. To do this we make use of a specific family of structual causal models—counterfactually independent models.

**Definition 14** (counterfactual independence (CFI)). *$Y$ is counterfactually independent in with respect to $A$ in $\mathcal{M}$ if,*

$$P(y_{a^*}^*, y_a|x) = \begin{cases} P(y_a|x)\delta(y_a - y_{a^*}^*) & a = a^* \\ P(y_{a^*}^*|x)P(y_a|x) & \text{otherwise} \end{cases} \tag{57}$$

Counterfactually independent models (CFI models) are those for which the outcome $Y_a$ is independent to any counterfactual outcome $Y_{a'}$. Next we show that there is always a CFI model that can induce any factual outcome statistics.

**Lemma 1.** *For any desired outcome distribution $P(y|a, x)$ there is a choice of exogenous noise distribution $P(e^Y)$ and causal mechanism $f_Y(a, x, e^Y)$ such that $Y$ is counterfactually independent with respect to $A$*

*Proof.* Consider the causal mechanism $y = f_Y(a, x, e^Y)$ for some fixed $X = x$, and exogenous noise distribution $P(E^Y = e^Y)$. Let the noise term by described by the random field $E^Y = \{E^Y(a, x) : a \in A, x \in X\}$, with $P(E^Y = e^y) = \times_{a \in A, x \in X} P(E^y(a, x) = e^y(a, x))$ and with $\text{dom}(E^Y(a, x)) = \text{dom}(Y)\,\forall\,A = a, X = x$. I.e. we choose the noise distribution to be joint state over mutually independent noise variables, one for every action $A = a$ and context $X = x$, and where each of these variables has the same domain as $Y$. Next, we choose the causal mechanism,

$$f_Y(a, x, e^Y) = e^Y(a, x) \tag{58}$$

i.e. the value of $Y$ for action $A = a$ and context $X = x$ is the state of the independent noise variable $E^Y(a, x)$. By construction this is a valid SCM, and we note that the factual distributions (calculated with (4)) are given simply by,

$$P(y|a, x) = P(E^Y(a, x) = y) \tag{59}$$

Likewise applying our choice of mechanism and noise distribution to (4) gives (for $a \neq a'$) the counterfactual distribution,

$$P(Y_a = y, Y_{a'} = y'|x) = P(E^Y(a, x) = y)P(E^Y(a', x) = y') \tag{60}$$
$$= P(Y_a = y|x)P(Y_{a'} = y'|x) \tag{61}$$

and likewise gives $P(y_a|x)\delta(y_a - y'_{a'})$ for $a = a'$. Finally, we note that we can choose any $P(y_a|x) = P(E^Y(a, x) = y)$, hence there is a CFI model that induces any factual outcome distribution we desire. $\qquad\square$

Next, we show that in counterfactually independent models there are outcome distributional shifts that only change the expected harm of individual actions, without changing any other factual or counterfactual statistics.

**Lemma 2.** *For $\mathcal{M}$ and (context-dependent) default action $A = \bar{a}(x)$, if $U$ is outcome dependent for the default action $\bar{a}(x)$ and some other action $a \neq \bar{a}(x)$, then there are three outcome distributionally shifted environments $\mathcal{M}_0$, $\mathcal{M}_+$ and $\mathcal{M}_-$ such that;*

*1. $\mathbb{E}[h|a, x; \mathcal{M}_-] < \mathbb{E}[h|a, x; \mathcal{M}_0] < \mathbb{E}[h|a, x; \mathcal{M}_+]$*

*2. $\mathbb{E}[h|b, x; \mathcal{M}_-] = \mathbb{E}[h|b, x; \mathcal{M}_0] = \mathbb{E}[h|b, x; \mathcal{M}_+] \, \forall \, b \neq a$*

*3. $P(y|a', x; \mathcal{M}_0) = P(y|a', x; \mathcal{M}_+) = P(y|a', x; \mathcal{M}_-) \, \forall \, a' \in A$, including $a, \bar{a}(x)$*

*Proof.* In the following we suppress the notation $\bar{a}(x) = \bar{a}$. To construct the environment $\mathcal{M}_0$ we restrict to a binary outcome distribution for each action such that $P(y_a|x)$ is completely concentrated on the highest and lowest utility outcomes,

$$Y_a = 1 \implies Y_a = \arg\max_y U(a, x, y) \tag{62}$$

$$Y_a = 0 \implies Y_a = \arg\min_y U(a, x, y) \tag{63}$$

$$1 = P(Y_a = 1|x; \mathcal{M}_0) + P(Y_a = 0|x; \mathcal{M}_0) \tag{64}$$

Note that we abuse notation as the variables $Y_a = 1$ and $Y_b = 1$ will not be in the same state in general, and the states $1, 0$ denote the max/min utility states under any given action, rather than a fixed state of $Y$. By Lemma 1 we can choose $Y_a$ to be counterfactually independent with respect to $A$. Recalling our parameterization of CFI models in Lemma 1, with noise distribuiton $P(E^Y = e^Y) = \times_{a \in A, x \in X} P(E^Y(a, x) = e^Y(a, x))$, $\text{dom}(E^Y(a, x)) = \text{dom}(Y)$, and causal mechanism $f_Y(a, x, e^Y) = e^Y(a, x)$, therefore $E^Y(a, x) \in \{0, 1\} \, \forall \, a, x$. The expected harm for action $\text{do}(A = a)$ is,

$$\mathbb{E}[h|a, x; \mathcal{M}_0] = \sum_{y_a=0}^{1} \sum_{y_{\bar{a}}=0}^{1} P(y_{\bar{a}}|x)P(y_a|x) \max\{0, U(\bar{a}, x, y_{\bar{a}}) - U(a, x, y_a)\} \tag{65}$$

where we have used the fact that $P(y_{\bar{a}}^*, y|a, x) = P(y_{\bar{a}}^*, y_a|x)$ and used counterfactual independence. $U(a, x, 0) < U(\bar{a}, x, 1)$ and so if we choose non-deterministic outcome distributions for $P(y_a|x)$ and $P(y_{\bar{a}}|x)$ then (65) is strictly greater than 0.

We can construct the desired $\mathcal{M}_{\pm}$ by keeping the causal mechanism but changing the factorized exogenous noise distribution in $\mathcal{M}$ to be,

$$P'(E^Y = e^Y; \mathcal{M}_+) = P(E^Y = e^Y; \mathcal{M}_0) + (-1)^{e^Y(a,x)-e^Y(\bar{a},x)}\phi_+ \tag{66}$$

$$P'(E^Y = e^Y; \mathcal{M}_-) = P(E^Y = e^Y; \mathcal{M}_0) + (-1)^{e^Y(a,x)-e^Y(\bar{a},x)}\phi_- \tag{67}$$

where $\phi_{\pm} \in \mathbb{R}$ are constants that satisfy the bounds $\max\{-P(Y_{\bar{a}} = 1|x)P(Y_a = 1|x), -P(Y_{\bar{a}} = 0|x)P(Y_a = 0|x)\} \leq \phi_{\pm} \leq \min\{P(Y_{\bar{a}} = 1|x)P(Y_a = 0|x), P(Y_{\bar{a}} = 0|x)P(Y_a = 1|x)\}$. It is simple to check that for any $\phi$ that satisfies these bounds we recover $\sum_{e^Y} P'(E^Y = e^Y) = 1$, $P'(E^Y = e^Y) \geq 0 \; \forall \, e^Y$, and therefore $P'$ is a valid noise distribution. Keeping the same causal mechanism $f_Y$ is $\mathcal{M}_{\pm}$ as in $\mathcal{M}_0$ gives $P(y_a|x; \mathcal{M}_0) = P(y_a|x; \mathcal{M}_+) = P(y_a|x; \mathcal{M}_-)$ as,

$$P'(y_i|x) = \sum_{e^Y(0,x)=0}^{1} \cdots \sum_{e^Y(i-1,x)=0}^{1} \sum_{e^Y(i+1,x)=0}^{1} \cdots \sum_{e^Y(|A|,x)=0}^{1} \left[ \prod_{j=1}^{|A|} P(e^Y(j,x)) + (-1)^{e^Y(i,x)-e^Y(\bar{a},x)}\phi_{\pm} \right] \tag{68}$$

$$= P(e^Y(i,x)) + (-1)^{e^Y(i,x)-0}\phi_{\pm} + (-1)^{e^Y(i,x)-1}\phi_{\pm} \tag{69}$$

$$= P(e^Y(i,x)) = P(y_i|x) \tag{70}$$

and likewise for $i = \bar{a}$. This implies that for any desired outcome statistics $P(y_a|x)$ there is a model where $Y_a \perp Y_{a'} \; \forall \, (a,a')$ where $a \neq a'$ except for the pair $a, \bar{a}$, so long as $P(y_{\bar{a}}|x)$ and $P(y_a|x)$ are non-deterministic (if they are deterministic, $\phi_{\pm} = 0$ and $\mathcal{M}_0 = \mathcal{M}_{\pm}$). Because $Y_{a'} \perp Y_{\bar{a}} \; \forall \, a' \neq a$, then $H(a',x; \mathcal{M}_0) = H(a',x; \mathcal{M}_{\pm}) \; \forall \, a' \neq a$. Also note that $H(\bar{a},x; \mathcal{M}) = 0$ for any $U$ or $\mathcal{M}$ if $P(a|x) = \delta(a - \bar{a})$, as $P(Y_{\bar{a}} = i, , Y_{\bar{a}} = k) = 0$ if $i \neq k$ and if $i = k$ (factual and counterfactual outcomes are identical) then the expected harm is zero. The only difference between $\mathcal{M}_0$ and $\mathcal{M}_{\pm}$ is $P(y_{\bar{a}}, y_a|x; \mathcal{M}_+) \neq P(y_{\bar{a}}, y_a|x; \mathcal{M}_-) \neq P(y_{\bar{a}}, y_a|x; \mathcal{M}_0)$, which differ for $\phi_+ \neq 0$, $\phi_- \neq 0$ and $\phi_+ \neq \phi_-$. Substituting (66) and (67) into our expression for the expected harm as using the notation $\Delta_{y,y'} = \max\{0, U(\bar{a},x,y) - U(a,x,y')\}$ gives,

$$\mathbb{E}[h|a,x; \mathcal{M}_{\pm}] = \mathbb{E}[h|a,x; \mathcal{M}_0] + \phi_{\pm}[\Delta_{00} + \Delta_{11} - \Delta_{10} - \Delta_{01}] \tag{71}$$

$$\mathbb{E}[h|a',x; \mathcal{M}_{\pm}] = \mathbb{E}[h|a',x; \mathcal{M}_0], \quad a' \neq a \tag{72}$$

Now, as $\max_y U(a,x,y) > \min_y U(\bar{a},x,y)$ then $\Delta_{01} = 0$. For the coefficient of $\phi_{\pm}$ in (71) to be zero, we would therefore require that $\Delta_{00} + \Delta_{11} = \Delta_{10}$. We know $\Delta_{10} > 0$ because otherwise $\min_y U(a,x,y) > \max_y U(\bar{a},x,y)$, therefore the minimal value of $\Delta_{10}$ is $\max_y U(\bar{a},x,1) - \min_y U(a,x,y)$. If $\Delta_{00} \neq 0$ and $\Delta_{11} \neq 0$ then $\Delta_{00} + \Delta_{11} \geq \Delta_{10}$ implies $\min_y U(\bar{a},x,y) \geq \max_y U(a,x,y)$ which violates our assumptions, therefore $\Delta_{00} + \Delta_{11} < \Delta_{10}$. If $\Delta_{00} = 0$ clearly we cannot have $\Delta_{11} = \Delta_{10}$ as $\min_y U(a,x,y) < \max_y U(a,x,y)$ by our assumptions, and likewise if $\Delta_{11} = 0$ we cannot have $\Delta_{00} = \Delta_{10}$ as this would imply $\min_y U(\bar{a},x,y) = \max_y U(\bar{a},x,y)$ which violates our assumptions. Therefore we can conclude that the coefficient in (71) in greater than zero.

Therefore if we choose any $0 < \phi_+ < \min\{P(Y_{\bar{a}} = 1|x)P(Y_a = 0|x), P(Y_{\bar{a}} = 0|x)P(Y_a = 1|x)\}$ we get $\mathbb{E}[h|a,x; \mathcal{M}_+] > \mathbb{E}[h|a,x; \mathcal{M}_0]$, and any $\max\{P(Y_{\bar{a}} = 1|x)P(Y_a = 1|x), P(Y_{\bar{a}} = 0|x)P(Y_a = 0|x)\} < \phi_- < 0$, we get $\mathbb{E}[h|a,x; \mathcal{M}_-] < \mathbb{E}[h|a,x; \mathcal{M}_0]$. $\qquad \square$

**Lemma 3.** *For (context dependent) default action $A = \bar{a}(x)$, $\mathbb{E}[h|\bar{a}(x),x; \mathcal{M}] = 0 \; \forall \, \mathcal{M}$*

*Proof.* In the following we suppress the notation $\bar{a}(x) = \bar{a}$.

$$\mathbb{E}[h|\bar{a}, x; \mathcal{M}] = \int_{y^*, y} P(Y_{\bar{a}} = y^*, Y = y|\bar{a}, x; \mathcal{M}) \max\{0, U(\bar{a}, x, y^*) - U(\bar{a}, x, y)\} \tag{73}$$

$$= \int_{y^*, y} P(Y_{\bar{a}} = y^*, Y_{\bar{a}} = y|x; \mathcal{M}) \max\{0, U(\bar{a}, x, y^*) - U(\bar{a}, x, y)\} \tag{74}$$

$$= \int_{y^*, y} P(Y_{\bar{a}} = y|x; \mathcal{M}) \delta(y^* - y) \max\{0, U(\bar{a}, x, y^*) - U(\bar{a}, x, y)\} \tag{75}$$

$$= \int_{y} P(Y_{\bar{a}} = y|x; \mathcal{M}) \max\{0, U(\bar{a}, x, y) - U(\bar{a}, x, y)\} \tag{76}$$

$$= 0 \tag{77}$$

$P(y|a, x) = P(y_a|x)$.

$\square$

**Theorem 3:** For any (context dependent) default action $A = \bar{a}(x)$, if there is a context $X = x$ where the user's utility function is outcome dependent for $\bar{a}(x)$ amd some other action $a \neq \bar{a}(x)$, then there is an outcome distributional shift such that $U$ is harmful in the shifted environment.

*Proof.* For the expected utility to not be harmful by Definition 6, it must be that $\mathbb{E}[h|a, x] > \mathbb{E}[h|b, x]$ $\implies$ $\mathbb{E}[U|a, x] < \mathbb{E}[U|b, x]$. Given our assumption of outcome dependence, we know there is a context $X = x$ such that the utility functions for $\bar{a}(x)$ and $a \neq \bar{a}(x)$ overlap, that is $\min_y U(a, x, y) < \max_y U(\bar{a}(x), x, y)$ and $\max_y U(a, x, y) > \min_y U(\bar{a}(x), x, y)$. In the following we drop the notation $\bar{a}(x) = \bar{a}$. We can restrict our agent to choose between these two actions and construct an outcome distributional shift such that; i) The outcomes $Y_a$ and $Y_{\bar{a}}$ are binary with one outcome maximizing the utility for that action and the other minimizing the utility, i.e. $Y_a \in \{\max_y U(a, x, y), \min_y U(a, x, y)\}$ and $Y_{\bar{a}} \in \{\max_y U(\bar{a}, x, y), \min_y U(\bar{a}, x, y)\}$, ii) $\mathbb{E}[U|a, x] = \mathbb{E}[U|\bar{a}, x]$, iii) $P(y_a|x)$ and $P(y_{\bar{a}}|x)$ are non-deterministic. This follows from the fact that the set of possible expected utility values for an action $a$ is the set of mixtures over $U(a, x, y)$ with respect to $y$, and as $Y_a = 0, 1$ are the extremal points of this convex set, the expected utility for action $a$ in context $x$ can be written as $P(Y_a = 0|x)U(a, x, 0) + P(Y_a = 1|x)U(a, x, 1)$. Then, as the utility functions for $a$ and $\bar{a}$ overlap there is point in the intersection of these convex sets that is non-extremal (and hence, a non-deterministic mixture).

By Lemma 3 the default action causes zero expected harm. By Lemma 2 we can construct a shifted environment $\mathcal{M}_0$ where the non-default action $a \neq \bar{a}$ has non-zero harm for any non-deterministic $P(y_a|x)$. We can therefore construct $\mathcal{M}_0$ such that i) $\mathbb{E}[Y_a|x] = \mathbb{E}[Y_{\bar{a}}|x]$, and ii) $\mathbb{E}[h|a, x] > \mathbb{E}[h|\bar{a}, x]$, violating our requirement that $\mathbb{E}[h|a, x] > \mathbb{E}[h|b, x] \implies \mathbb{E}[U|a, x] < \mathbb{E}[U|b, x]$.

$\square$

**Theorem 4:** For any (context dependent) default action $A = \bar{a}(x)$, if there is a context $X = x$ where the user's utility function is outcome dependent for $\bar{a}(x)$ and two other actions $a_1, a_2 \neq \bar{a}(x)$, then for any factual objective function $J$ there is an outcome distributional shift such that maximizing the $J$ is harmful in the shifted environment.

*Proof.* By assumption there is a context $X = x$ for which the utility functions for $a_1, a_2$ and $\bar{a}(x)$ overlap. In the following we drop the notation $\bar{a}(x) = \bar{a}$. There is a choice of non-deterministic outcome distributions $P(y_{\bar{a}}|x)$, $P(y_{a_1}|x)$ and $P(y_{a_2}|x)$ such that all three actions have the same expected utility. By Lemma 2 for any non-deterministic outcome distribution we can choose $\mathcal{M}_0$ such that $\mathbb{E}[h|a_1, x; \mathcal{M}_0] > 0$, and $\mathbb{E}[h|a_2, x; \mathcal{M}_0] > 0$, and by Lemma 3 $E[h|\bar{a}, x; \mathcal{M}] = 0 \,\forall\, \mathcal{M}$. Therefore $\exists \,\mathcal{M}_0$ that is an outcome distributional shift of the original environment $\mathcal{M}$ such that $\bar{a}, a_1, a_2$ have the same expected utility, $\bar{a}$ has zero expected harm and $a_1, a_2$ have non-zero expected harm.

If $\mathbb{E}[h|a_1, x\mathcal{M}_0] = \mathbb{E}[h|a_2, x; \mathcal{M}_0]$ then by Lemma 2 there are outcome-shifted environments $\mathcal{M}_\pm$ such that $\bar{a}, a_1$ and $a_2$ have the same factual statistics as in $\mathcal{M}_0$ and $\mathbb{E}[h|a_2, x\mathcal{M}_0] = \mathbb{E}[h|a_2, x\mathcal{M}_\pm]$, but the harm caused by $a_1$ is increased(descreased) by some non-zero amount. Therefore in $\mathcal{M}_+$ $a_1$ and $a_2$ have the same expected utility but $a$ has a strictly higher expected harm, and in order to be non-harmful it must be that $\mathbb{E}[J|a_1, x; \mathcal{M}_+] < \mathbb{E}[J|a_2, x; \mathcal{M}_+]$. Likewise in $\mathcal{M}_-$ $a_1$ and $a_2$ have the same expected utility but the expected harm for $a_1$ is strictly lower than for $a_2$, therefore in order to be non-harmful it must be that $\mathbb{E}[J|a_1, x; \mathcal{M}_-] > \mathbb{E}[J|a_2, x; \mathcal{M}_-]$. Finally we note that $\mathbb{E}[J|a, x; \mathcal{M}_+] = \mathbb{E}[J|a, x; \mathcal{M}_-] = \mathbb{E}[J|a, x; \mathcal{M}_0] \ \forall \ a \in A$ as the factual statistics are identical in $\mathcal{M}_0, \mathcal{M}_\pm$, i.e. $P(y_a|x; \mathcal{M}_+) = P(y_a|x; \mathcal{M}_-) = P(y_a|x; \mathcal{M}_0)$. Therefore any $J$ must be harmful in either $\mathcal{M}_+$ and $\mathcal{M}_-$, and therefore there is an outcome distributional shift $\mathcal{M} \to \mathcal{M}_+$ or $\mathcal{M} \to \mathcal{M}_-$ such that $J$ is harmful in the shifted environment.

If $\mathbb{E}[h|a_1, x\mathcal{M}_0] \neq \mathbb{E}[h|a_2, x; \mathcal{M}_0]$, assume without loss of generality that $\mathbb{E}[h|a_1, x\mathcal{M}_0] > \mathbb{E}[h|a_2, x; \mathcal{M}_0]$. As $\bar{a}, a_1$ and $a_2$ have the equal expected utilities then so does any mixture of these actions, in $\mathcal{M}_0$ and $\mathcal{M}_\pm$. Restrict the agent to choose between action $a_2$ and a mixture of actions $\bar{a}$ and $a_1$—i.e. a stochastic or 'soft' intervention [19, 85], which involves replacing the causal mechanism for $A$ with a mixture $\tau := q[A = a_1] + (1 - q)[A = a_0]$ where $q$ is an independent binary noise term. By linearity the expected utility for this mixed action is $\mathbb{E}[U_\tau|x] = q\mathbb{E}[U_{a_1}|x] + (1-q)E[U_{\bar{a}}|x] = \mathbb{E}[U_{a_1}|x]$ as all three actions have the same expected utility, and has an expected harm $\mathbb{E}[h|\tau, x; \mathcal{M}_0] = q\mathbb{E}[h|a_1, x; \mathcal{M}] + (1-q)\mathbb{E}[h|\bar{a}, x; \mathcal{M}] = q\mathbb{E}[h|a_1, x; \mathcal{M}]$ as $\mathbb{E}[h|\bar{a}, x; \mathcal{M}] = 0 \ \forall \ \mathcal{M}$. Therefore as $\mathbb{E}[h|a_1, x; \mathcal{M}] > 0$ and $\mathbb{E}[h|a_2, x; \mathcal{M}] > 0$ we can choose $p > 0$ such that $\mathbb{E}[h|\tau, x; \mathcal{M}_0] = \mathbb{E}[h|a_2, x; \mathcal{M}_0]$. Therefore in $\mathcal{M}_0$, $a_2$ and $\tau$ have the same expected harm and utility, and in $\mathcal{M}_+$ they have the same expected utility but $\tau$ is more harmful than $a_2$ as $\mathbb{E}[h|a_1, x; \mathcal{M}_+] > \mathbb{E}[h|a_1, x; \mathcal{M}_0]$ and $p > 0$, and in $\mathcal{M}_-$ they have the same expected utility but $a_2$ is more harmful that $\tau$. As the factual statistics $P(y_a|x)$ are identical for $\mathcal{M}_0$ and $\mathcal{M}_\pm$, so is the value of any factual objective function across all three environments. Hence, any factual objective function must be harmful in either $\mathcal{M}_+$ or $\mathcal{M}_-$. $\qquad\square$

# L

In this appendix we discuss the limitations of our results, explore some potential applications to more complicated domains, and discuss related works; namely counterfactual fairness [33] and path-specific objectives [75].

## L.1   Limitations

Our proposed framework for dealing with harm does not come without limitations to be investigated in future work. Similar to other works on causal inference (see [62] for review), our current setup assumes that all variables are observed when computing the counterfactual (no unobserved confounding), which may limit the applicability of our measure for harm to certain scenarios. For the sake of generality our theoretical results are derived in the SCM framework, and so taken at face value they assume knowledge of the SCM for the data generating process. While Theorem 4 establishes that there is no factual objective function that can give robust harm aversion in all situations, calculating the expected harm using counterfactual inference is not always the most appropriate method in some cases (such as when the SCM or some good approximation of it is not known). Several methods have been proposed for results with similar restrictions (e.g. in counterfactual fairness [95]). One approach that is particularly appropriate for dealing with harm is bounding counterfactuals, which allows for tight upper bounds to equation (6) to be dervied using knowledge of the interventional distributions [25] or a combination of observational data, interventional data, and assumptions on the generative functions (such as monotonicity) [96]. Using this upper bound in place of the counterfactual harm in the HPU (Definition 4) is sufficient to ensure that the desired degree of harm aversion is satisfied (guaranteeing no actions violate a fixed harm-benefit ratio). One example of this upper bound is for binary actions and outcomes, i.e. for $A, Y \in \{0, 1\}$ where without loss of generality $\bar{a} = 0$, and $U = U(a, y)$. Using the tight bounds from [25],

$$P(Y_0 = 1, Y_1 = 0) \leq \min\{P(Y_0 = 1), P(Y_1 = 0)\} \tag{78}$$

Note that the RHS of this inequality is identifiable without knowledge of the SCM. From this we can upper bound the harm,

$$h(A = 1, Y = 0) = \sum_{y^* \in \{0,1\}} P(Y_0 = y^* | y, a) \max\{0, U(0, y^*) - U(a, y)\} \tag{79}$$

$$\leq \sum_{y^* \in \{0,1\}} \min\{1, P(Y_1 = 0)/P(Y_0 = y^*)\} \max\{0, U(0, y^*) - U(1, 0)\} \tag{80}$$

$$tw \tag{81}$$

Another potential limitation is the assumption of knowledge of the SCM (or a good approximation), which is unlikely to be valid in complex domains. In section 6 we demonstrated our theoretical results in the relatively simple domain of dose-response analysis using GAM models. On the one hand these settings can be of high impact (for example GAMs are arguably the most widely used models in clinical trials, epidemiology and social policy precisely because they are simple and interpretable, and these domains have a high risk of harm from misspecified models [97]). This also allows us to observe that harm aversion has a significant impact even in these simple scenarios, without these results potentially being due to the intricacies of specific model architectures or inference schemes. However, in future work our framework should be extendable to more complex domains where we do not typically have access to the true SCM that describes the data generating process but some approximation (one exception being in simulations, where counterfactuals can be directly measured as the exogenous noise state is known).

There have been several recent proposals for performing counterfactual inference using deep learning methods, with promising results in diverse complex domains including learning deep structural causal models for medical imaging [37], visual question answering [38], vision-and-language navigation in robotics [39] and text generation [40]. These studies evidence that deep learning algorithms can learn to make good counterfactual inferences that can be used to support decision making without perfect knowledge of the underlying SCM. This is somewhat analogous to the fact that human decision making often utilizes counterfactual reasoning for various cognitive tasks [98] (for example, it is important for legal and ethical reasoning [28]), in spite of the fact that humans clearly do not having access to perfect structural causal models of their environments but learn approximations through heuristics and inductive biases [99].

We now briefly discuss two examples of current implementations of counterfactual reasoning in complex domains where our framework could be applied.

**Example: medical imaging.** Consider a recent study developing a deep structural causal models for generating counterfactual images of brain CT scans in patients with multiple sclerosis (MS) [100]. MS causes brain lesions and abnormalities, and CT scans of the brain can be used to predict health outcomes for patients including disease progression and long-term patient outcomes [101]. MS is known to have a wide range of demographic risk factors including smoking and exposure to chemical pollutants, and these factors are known to cause artefacts in brain scans such as lesions independently of MS. To determine the harm caused by the patient by the disease one has to determine what the patient's 'healthy' scan would look like (if they did not have MS), given their factual scans which encode information about latent factors such as smoking and exposure to environmental pollutants (which also cause negative health outcomes through causal pathways that are not mediated by MS). Translating to our framework, the counterfactual harm Definition 3 can be estimated by generating samples of counterfactual healthy images $y^* \sim P(Y_{\bar{a}} | a, y; \mathcal{M})$ where $Y_{\bar{a}}$ is the counterfactual image under the intervention that sets the latent variable for duration of symptoms to zero (as in used to generate healthy images in [100]), $Y$ is the factual image, $A = a$ is the known factual duration of symptoms and $\mathcal{M}$ is the deep structural causal model derived in [100]. Finally, the counterfactual harm (6) can be estimated using a reasonable utility function $U(y)$ such as the predicted quality adjusted life years (QALYs) for a given CT scan evaluated using a deep learning model for predicting patient outcomes [101]. The resulting harm measure would be the expected decrease in QALYs caused by the presence of MS.

**Example: reinforcement learning.** Consider a recent study where a reinforcement learning agent is trained to determine optimal treatment policies for major depression [102]. A structural causal model is learned for the Markov decision process and used to generate counterfactual explanations for patient outcomes. Sequential decision making in medicine is a good use-case for our framework as there are well defined default treatment policies $\pi(\bar{a}|s)$ (e.g patients with certain medical history receive a standardized treatments), and there is increasing interest in using reinforcement learning

techniques to improve patient outcomes by adapting treatments over time and personalizing them to patients (see for example [103, 104]). However, care must be taken to ensure that any learned treatment policies are not overly harmful compared to the standardized treatment policies that the patient would have received. For example, some treatment policies may improve patient outcomes on average by benefiting some patients while causing other patients worse outcomes than they would have had (much like with the allergic reaction in Examples 1 & 2—indeed these heterogeneous responses to treatments are commonplace in psychiatry and medicine in general). To apply our framework for harm aversion the agent can be trained with a harm-averse Bellman equation,

$$Q_\lambda(a, s; \mathcal{M}) = \sum_{s'} P(s'|a, s) \left[ R(a, s, s') - \lambda h(a, s, s'; \mathcal{M}) + \gamma \sum_{a'} \pi(a'|s') Q_\lambda(a', s'; \mathcal{M}) \right]$$
(82)

where $\mathcal{M}$ is an SCM of the Markov decision process (derived in [102]) and,

$$h(a, s, s'; \mathcal{M}) = \sum_{s^*} P(S'_{\bar{a}(s)} = s^*|a, s, s') \max\{0, R(\bar{a}(s), s, s^*) - R(a, s, s')\}$$
(83)

where $a(s)$ is the deterministic default treatment choice that the patient would receive if they followed the standardized treatment rules. For $\lambda = 0$ this reduces to the standard Bellman equation, but for $\lambda > 0$ the agent chooses a policy that maximizes the discounted cumulative HPU (Definition 4) rather than the reward, and so will avoid actions that achieve higher cumulative reward at a harm-benefit ratio less than $1 : 1 + \lambda$ (as described in section 4 and illustrated in Section 6).

### L.2 Relation to bigger picture AI safety and ethics

There very real potential for algorithms to cause harm have been widely studied (see for example [105, 106, 107, 108, 109] for reviews). It is important to note that our results do not solve these established AI safety problems. Harm-averse agents will still be susceptible to issues such as failures of robustness, poor risk aversion, model misspecification, and unsafe exploration, which can ultimately lead to harmful outcomes even if the agent is harm averse. For example, the problem of model-misspecification has been studied in relation to attempts to deal with algorithmic bias using causal models [95], and we expect harm-averse agents to be similarly susceptible to problems of model misspecification. Instead, our results point out a new failure mode in addition to these known examples—that algorithms that pursue factual objectives (which describes the vast majority of existing implementations) are incapable of robustly reasoning about harm and avoiding harmful actions. The solution we propose is to motivate the design of algorithms that are capable of reasoning counterfactually about their actions and objectives.

### L.3 Related work

In its original conception counterfactual fairness deals with prediction tasks $\hat{Y} : X \to Y$ where the desire is to have a predictor $\hat{Y}$ that is not unfairly influenced by a protected attribute $A$ such as gender or race. Note $A$ is a feature that typically cannot be intervened on, whereas is our setup $A$ denotes an agent's action. Harm is conceptually distinct from fairness—for example, it is possible to apply a strictly harmful action fairly—but the two measures can be used in tandem. For example, one could quantify if a action or decision was unfair, and whether or not the user was harmed due to this unfair action. Counterfactual fairness quantifies this unfair influence causally, using the counterfactual constraint,

$$P(\hat{Y}_a = y|X = x, A = a) = P(\hat{Y}_{a'} = y|X = x, A = a) \quad \forall a' \in A, y \in \hat{Y}$$
(84)

which states that the probability of predicting any given outcome should not be caused on average by the protected attribute $A$, where the counterfactual $P(\hat{Y}_{a'} = y|X = x, A = a)$ is the probability of $\hat{Y}$ given $A = a$ if $A$ had been equal to $a'$. Note that the counterfactual in (84) does not deal with the joint statistics of the factual outcome $\hat{Y}_a$ and the counterfactual outcome $\hat{Y}_{a'}$. In [74] a path-specific variant of counterfactual fairness in proposed, which similar to our results makes use of counterfactual

contrasts with a baseline (in the case of [74] this a baseline value of the sensitive attribute). Similar to path-specific objectives below, this approach consider the case where there are fair and unfair causal pathways for the sensitive attribute to influence $Y$ (where $Y$ is a decision variable).

Another perhaps more related use of counterfactual inference for ethical AI is path-specific objectives [75]. This work refines the expected utility to take into account the fact that we often want to maximize utility via specific causal pathways due to ethical constraints. For example we can consider a simple model where the agent's action $A$ influences user feedback $Y$ (and utility $U(y)$) but also effects the users preferences $H$ where $A \rightarrow Y$, $A \rightarrow H$ and $H \rightarrow Y$. To maximize utility without intentionally manipulating the user we must maximize along the causal pathway $(1) : A \rightarrow Y$ without including contributions to the expected utility from the mediator pathway $(2) : A \rightarrow H \rightarrow Y$. This involves replacing the expected utility with its path-specific counterfactual equivalent, much in the same way that our path-specific harm (Definition 9) generalizes our path-independent definition of harm (Definition 3). As such the path-specific expected utility is still agnostic to harm just as the expected utility is, although it could be combined with the path specific harm in [75] to give a path-specific variant of the HPU (Definition 4). This would allow for harm averse decision making where the necessary degree of harm-aversion $\lambda_{(i)}$ varies depending on the causal path $(i)$—for example, if we desire agents that have a high aversion for being directly harmful, but a lower degree of harm-aversion for indirect harm mediated by the actions of other agents (as described in Appendix B).