# OpenReview forum: "Counterfactual harm"
_NeurIPS.cc/2022/Conference — NeurIPS 2022 Accept_

### Official Review · Reviewer_tHHv · 2022-07-04

**Rating:** 7
**Confidence:** 4
**Soundness:** 3 good
**Presentation:** 3 good
**Contribution:** 2 fair

**Summary:**

The paper argues that harm, evasion of which is key for AI to be safe and ethical when deployed, is intrinsically a counterfactual quantity. Further, that conventional ML is guaranteed to end up with harmful policies if unable to perform counterfactual reasoning. As contribution, it claims to be the first to produce a statistical definition of harm, and have derived a set of counterfactual objective functions to robustly mitigate harm. Finally, using a statistical model for indentifying optimal drug doses, the paper demonstrates its framework that integrates harm into algorithmic decisions can dentify doses that are significantly less harmful while not sacrificing efficacy whilst standard algorithms yield significant harm.

**Questions:**

If indeed tha paper elucidated or proved that the proposed counterfactual-based harm-aversed framework can account for more complex harmful situations, it would be good for the authors to make this explicit. The paper also showed how outcome-dependent optimal policy learning happens within the framework. How about the framework being able to learn amid contextual, P(x; M), changes? In otherwords, when distribution shift is not just on the outcome distribution, but also on the input (e.g., covariate shifts) or to both (concept drift)?  Lastly, some support (proof) to qualify certain claims (assumptions), as follows:
1. The  claim that agents trained to maximize factual objective functions are guaranteed to proceed with harmful policies, albeit they are allowed to retrain, amid distributional shifts (L332~L334). For one, Scholkopf et al. [1] argues that change in data distribution "may lead to (arbitrarily) in accurate predictions". Thus, this claim should be situated on the premise that the characterized objective function is pertinent only to domains where there is vulnerability to harm. Secondly, however, why would this claim hold even if the agent can retrain itself?
2. Why was the requirement that agents have some fixed harm aversion relaxed, and tapered to requiring only that the agents do not take needlessly harmful actions?

**Limitations:**

On the two brief mentions on limitation, namely, (1) that it remains "an open question as to how counterfactual reasoning can be achieved with current implementations", and (2) "there is growing interest in counterfactual reasoning with deep learning models", it would help to discuss this further in a section on related works.

**Strengths And Weaknesses:**

The paper is very well written. The social significance of the research in general, together with its arguments, research questions and formal elucidations, are well positioned and articulated.

However, inherent in formal frameworks and models is to abstract or simplify potentially required strong ties to more complex real-world contexts whilst claim generalizability. While a generalized additive model was used to illustrate the counterfactual framework for dose-response predictions, how to scale up to a rather complex domain (e.g., hazardous environment in which human and AI teamp up to navigate the situation) is worth discussing. Further, it is unclear how the causal constructs elucidated here accounted for or will account for human feedback, e.g., Alice's preferences, clinician's sense of values or human demonstrations in general (objective functions, which for example provably beneficial AI methods account for).  While it is not a question for me that this is the first to  provide a statistical defintion of harm and render  it a counterfactual framework, the discussion on related works seems to miss the built up on why the current literature has not addressed it until this time, whether previous works have sought or alluded this reasearch direction and might be falling short, and how close (or far) current deep learning frameworks on counterfactual reasoning relate to this work.

---

> ### Author Response · Authors · 2022-08-01
> **Response to reviewer tHHv**
>
> We thank the reviewer for their thoughtful feedback.
>
> Q1. How to scale up to a rather complex domain (e.g., hazardous environment in which human and AI teamp up to navigate the situation) is worth discussing.
>
> We agree this point was missing from the paper and we have included a new appendix K to discuss this. The example you give is described in recent paper that we discuss [1].
>
> [1] Parvaneh, Amin, et al. "Counterfactual vision-and-language navigation: Unravelling the unseen." Neurips 2020.
>
>
> Q2. the discussion on related works seems to miss why the current literature has not addressed (harm) until this time, whether previous works have...falling short, and how close (or far) current deep learning frameworks on counterfactual reasoning relate to this work.
>
> We agree this is lacking and have included in appendix K (and discuss alternatie definitions that fail in appendix B)
>
> Q3. Further, it is unclear how the causal constructs elucidated here accounted for or will account for human feedback, e.g., Alice's preferences, clinician's sense of values or human demonstrations in general (objective functions, which for example provably beneficial AI methods account for).
>
> We discuss this briefly in example 4 and the paragraph beginning line 319. As cooperative inverse reinforcement learning and RL from human feedback use factual reward models, they cannot be robustly harm averse. consider the treatment example from the introduction. An agent can be trained on feedback that treatment 1 is good. Then a distributional shift can result in treatment 1 behaving like treatment 2, without any of the factual statistics changing. It will still maximize the factual reward.
>
>
> Q4. If indeed...counterfactual-based harm-averse framework can account for more complex harmful situations, it would be good for the authors to make this explicit.
>
> We have included a note on this in Appendix K, describing existing work on counterfactual inference in complex domains where our definitions should be applicable. Our no-go theorems (theorems 3 and 4) apply to these more complicated scenarios (for example, if harm aversion is impossible in single decision tasks it is by extension impossible in MDPs).
>
>
> Q5. The claim that agents trained to maximize factual objective functions are guaranteed to proceed with harmful policies, albeit they are allowed to retrain, amid distributional shifts (L332~L334).... Secondly, however, why would this claim hold even if the agent can retrain itself?
>
> Theorems 3 & 4 are different to what you describe here. What we show is that for any factual objective function, there exist environments where maximising this objective functions results in needlessly harmful actions. For example, consider an agent that minimizes the L2 norm in all environments. Maybe this results is low harm actions in the training environment, and the designers conclude `the L2 norm is a good harm averse objective’. However, we prove that there are shifted environments where minimising the L2 norm results in needlessly harmful actions. This is not due to generalisation error, but “objective misgeneralization”. By allowing to retrain, we make the generalisation error zero so as to make this clear. Consider again the treatment example from the introduction. Let in the training environment both treatments behave like treatment 1 (e.g. f_Y(t=2, e_Y) -> f_Y(t=1, e_Y), so no harm). In this environment, both treatments cause not harm and any factual objective (e.g. maximising the treatment effect) will be harm averse. Then imagine a distributional shift where now f_Y(t=2, e^Y) returns to its original (harmful) form. Note the factual outcome statistics do not change under this shift. So T = 2 will still be optimal w.r.t the treatment effect (or any other factual measure), but is now harmful. So a needlessly harmful action belongs to the optimal set. The theorem shows this is true in general.
>
>
> Q6. How about the framework being able to learn amid contextual, P(x; M), changes? In otherwords, when distribution shift is not just on the outcome distribution, but also on the input (e.g., covariate shifts) or to both (concept drift)?
>
> As we allow the agent to re-train following the distributional shift, covariate drive and concept drift are not relevant as the outcome distribution is essentially known by the agent following the distributional shift, and this is the only distribution required to determine the optimal policy (e.g. once P(y | a, x) is known, the optimal policy is independent of P(x)).
>
> Q7. Why was the requirement that agents have some fixed harm aversion relaxed, and tapered to requiring only that the agents do not take needlessly harmful actions?
>
> This is to strengthen the theorem. Otherwise, maybe there are heuristics for approximating harm using factual statistics that get it approximately right (resulting in some approximation of the desired harm aversion). The theorem proves this is impossible

---

> > ### Comment · Reviewer_tHHv · 2022-08-08
> > **Further from the Author's response**
> >
> > Thank you very much for addressing the comments. Explaining further indeed helped to clarify.  On Q1 and Q2, however, after reading the added Appendix K, perhaps, further, albeit brief, explanation would help. While it is indeed the case that we do not (and may not at all) have access to the perfect causal relations in complex domains and, thus, can only rely on approximations, explicitly, how does this connect to the proposed model? Using for instance from the simple treatment-outcome set-up with binary values to more a complex scenario as hazard management.

---

> > > ### Author Response · Authors · 2022-08-08
> > > **Response 2 to reviewer tHHv**
> > >
> > > We would like to really thank the reviewer for their comments. We agree that the paper would be improved by giving a more specific explanation of how our formalism could fit into these more complex domains. Following their recommendations we have significantly expanded appendix K to include two examples of recent works where our framework can be applied to complex domains, including details as to how the counterfactual harm can be calculated in these settings.
> > >
> > > The first example deals with complex state spaces (medical imaging). We describe how the counterfactual harm can be estimated in a recent study looking at counterfactual inference for CT scans of patients with multiple sclerosis
> > >
> > > The second example looks at another axis of complexity---sequential decisions. We describe how our harm averse decision making (section 4) can be applied a recent paper on counterfactual inference in Markov decision processes. This study looks at determining optimal treatment policies for patients with major depression. We describe how harm averse policies can be learned using an extension of the Bellman equation that factors in the expected harm.
> > >
> > > We have also included a section in appendix K describing related work in fairness and AI safety / ethics

---

> > > > ### Comment · Reviewer_tHHv · 2022-08-10
> > > > **Comments addressed**
> > > >
> > > > Thank you for addressing the comments and extending the submitted materials. The paper has been made more comprehensible.

---

> > > > > ### Comment · Reviewer_tHHv · 2022-08-10
> > > > > **Updated ratings**
> > > > >
> > > > > I revised my assessment of the work based on the answers provided by the authors.

---

### Official Review · Reviewer_8Y89 · 2022-07-04

**Rating:** 6
**Confidence:** 4
**Ethics Flag:** Yes
**Soundness:** 3 good
**Presentation:** 1 poor
**Contribution:** 2 fair

**Summary:**

This paper rigorously define the concept of "counterfactual harm" using statistical analysis. Overall, the authors quantify the "benefit" & "harm" based on a common definition of counterfactual comparative account (CCA). With the mathematical formulation of counterfactual harm, the authors further derive a family of counterfactual objective functions for mitigation. Solid theoretical analysis are provided with the structural causal modeling, and an empirical example illustration is shown to demonstrate the effectiveness of the proposed quantification of counterfactual harm.

**Questions:**

1. I'm wondering how we can apply this definition (harm quantification) to a general ML setting, so as to mitigate some potential biases learned in models? Is the SCM a necessary setting for incorporating the proposed family of objective functions?

2. In a more specific setting (e.g., fairness domain), can we equate or connect the harm quantification to some existing measurements (e.g., counterfactual fairness - Kusner, Matt J., et al. "Counterfactual fairness." NeurIPS'17)? If yes, then what is the key difference for the proposed harm quantification?

3. Is it possible to design some validation experiments on real-world data for this work?

**Ethics Review Area:**

["Inappropriate Potential Applications & Impact  (e.g., human rights concerns)"]

**Limitations:**

1. The major theoretical analysis & results are purely based on the statistical causal analysis, which may not fit in the settings of standard ML scenarios. Personally, I think the proposed work could be limited since it is hard to directly employ such harm quantification to mitigate the "harmful" patterns existing in commonly used models, such as neural nets.

2. The effectiveness of this work on wider domains still needs further validation, especially with the experiments on real-world datasets. Current illustration part could be limited to fully evaluate the proposed harm measurements.

3. The authors should conduct a more detailed discussion between this work and other existing frameworks for quantifying "harmful" concepts (e.g., fairness, privacy). It seems like the authors try to make the definition from a higher level, but we readers/researchers also need to know what is the generality brought by this paper. In the current version, this part is quite limited in my view.

**Strengths And Weaknesses:**

Strengths:

- The topic of this paper is important and significant, which can largely benefit the areas such as fairness AI & ethical AI. The provided quantification method is general, so many down-stream application scenarios could refer.

- The equations in the main body of the paper is well introduced and clarified, which is easy to follow.

- The authors provide many contexts and background information which is very useful for readers. The provided high-level examples can help quickly extract the main idea of this paper.

Weakness:

- The overall paper is not well structured. The section titles are kind of unclear.

- The analysis setups are too specific, even though the harm definition is general by itself. This could be limited when applying such quantification scheme to a general ML settings.

- The paper only provides a simple illustration for demonstration, without any real-world experiments & comparison. This makes the proposed work hard to be fully evaluated regarding to its overall performance.

---

> ### Author Response · Authors · 2022-08-01
> **Response to reviewer 8Y89**
>
> We thank the reviewer for their thoughtful feedback and are encouraged by their description of the the topic of the paper being “important and significant” and our approach as being “rigorous” and “general”. We address their comments and questions below.
>
> Q1 The paper only provides a simple illustration for demonstration, without any real-world experiments & comparison.
> * The effectiveness of this work on wider domains still needs further validation, especially with the experiments on real-world datasets.
>
> Its not correct that we do not provide experiments using real-world data. The illustration in section 6 uses a GAM model that is learned on real world experimental data (a meta-analysis for randomised control trials), and we have made this clearer in the text. By comparison, in the neurips paper “counterfactual fairness” which the reviewer compares our work, the authors of this paper illustrate their definition of fairness on a fictitious model with purely synthetic data, whereas our results are demonstrated on a model learned on real data and which is used in practice to determine optimal doses of the drug Ariprazole. We chose a simple demonstration deliberately, as it shows that even in simple settings harm aversion results in very different policies to harm-indifferent optimisers. In more complex demonstrations it would be less clear if this effect is due to the specifics of the model architecture. (for more complex implementations see appendix K)
>
>
> Q2 The analysis setups are too specific, even though the harm definition is general by itself. This could be limited when applying such quantification scheme to a general ML settings.
> * Personally, I think the proposed work could be limited since it is hard to directly employ such harm quantification to mitigate the "harmful" patterns existing in commonly used models, such as neural nets.
>
> Following your recommendations we have now included an Appendix K which details how our framework could be applied to several recent setups where counterfactual inference is employed in complex real-world settings (e.g. using deep structural causal models). This includes in sequential decision making with an RL example, and in medical images.
>
> While our main demonstrations uses GAMs and these are not deep learning models, they are the most common models used for decision support systems in areas where our results will have most impact---namely in clinical trials, epidemiology, and social policy. These models are used in these fields precisely because they are simple and interpretable, and the risk of harm from model misspecification for complicated models is so high
>
> Finally, there is arguably a catch 22 problem here—The main reasons for developing deep learning systems that can support counterfactual inferences are given by theoretical results such as ours that show counterfactual inferences are necessary for practical problems (like harm aversion).
>
> Q3. I’m wondering how we can apply this definition (harm quantification) to a general ML setting, so as to mitigate some potential biases learned in models? Is the SCM a necessary setting for incorporating the proposed family of objective functions?
>
> We use SCM models to derive our theoretical results as is common practice in Pearlean causality. For example the do calculus is derived in the SCM formalism but is applied to a widely, including deep generative models. We discuss in appendix K the potential limitations for this approach in practice, and how these are being overcome in several recent works that approximate SCMs and counterfactual inference without having to know the true underlying SCM. This includes deep learning methods and bounds on counterfactuals.
>
>
> Q4. In a more specific setting (e.g., fairness domain), can we equate or connect the harm quantification to some existing measurements (e.g., counterfactual fairness -"Counterfactual fairness."? If yes, then what is the key difference for the proposed harm quantification?
>
> We have included a discussion of the relation between counterfactual fairness and harm in appendix K. Briefly, these measures are different both technically (type vs actual causality, fairness is about estimator for outcome Y rather than outcome itself). On a conceptual level harm is distinct from fairness. For example, a treatment could harmful but administered in a fair way that doesnt discriminate based on a protected attribute. We discuss how they can be used in tandem (e.g. identifying if harm has been caused by an unfair decision).
>
> Q5. The authors should conduct a more detailed discussion between this work and other existing frameworks for quantifying "harmful" concepts (e.g., fairness, privacy).
>
> Following the reviewers suggestion we have included a discussion of this in appendix K.

---

> ### Author Response · Authors · 2022-08-09
> **Follow up with the rebuttal**
>
> Dear Reviewer 8Y89
>
> The discussion period is coming towards its end. We wonder whether you had the chance to check our rebuttal and see if it clarified the interesting issues raised in your review (as we hoped). Otherwise, we will be happy to follow up and provide further elaboration on unanswered concerns and burning questions.
>
> We certainly appreciate your time and attention. Thank you!

---

> > ### Comment · Reviewer_8Y89 · 2022-08-09
> > **Increase my score**
> >
> > Thanks the authors for the very detailed rebuttal.
> >
> > By reading the clarifications provided by authors, as well as those helpful discussions among other reviewers, I misunderstood part of this paper. I think the authors' rebuttal convince me, so I would like to increase my score to 6.
> >
> > Thanks again for the explanation.

---

> > > ### Author Response · Authors · 2022-08-09
> > > **Thank you**
> > >
> > > Massive thank you for your comments which have really improved the paper with the inclusion of the related works discussion. Also many thanks for considering the reviews and revising your score.

---

### Official Review · Reviewer_rBDL · 2022-07-11

**Rating:** 6
**Confidence:** 4
**Soundness:** 3 good
**Presentation:** 3 good
**Contribution:** 2 fair

**Summary:**

This paper considers defining harm as a counterfactual quantity involving user’s utility. The harm is defined based on the expected difference in the utility following the policy and the utility following the default action. Equipped with the definition of harm, the authors proposed harm penalized utility [HPU] and relate to its implications to decision making. Authors investigated a few properties of the definition(s) and provide a proof-of-concept simulation result in the context of dose-response showing that a policy can be optimized to lower harm depending on the user specified hyper-parameter.


**Questions:**


- In the example in the introduction, I understand that why doctors would favor Treatment 1 vs 2. This feels so much like a trolley problem where inaction can cause 5 people die and action (switching) can cause 1 people die. What’s your explanation? In HPU, U is -5, -1 and harm is -1. So with lambda 4, there is no difference between switching lanes?
- Def 1, why should noise be mutually independent?
- Is utility fixed over the entire population? A is the agent's action but U is user's utility function. User != Agent. (Y would also be user's outcome...)
- What's the practical importance of Y in defining harm and benefit when you have utility U? Y exists only to be marginalized out. For decision making, how would you use Y? where Y is not observed for the current user and X is given...

**Ethics Review Area:**

["I don’t know"]

**Limitations:**

Please see the weaknesses especially about the non-identifiability of the measure. One may show the applicability of new definition under a learned model, etc.
=======
I raise my score which now reflects better understanding of the paper.

**Strengths And Weaknesses:**

I would like to thank the authors first for an intriguing notion for harm (and benefit).

Strengths
- As a causal inference researcher, seeing the definition of harm at the level of “counterfactual” not “intervention” is refreshing.

Weaknesses
- It is unclear why we should define harm and benefit separately by taking max(0, difference in utility)
- Simply the P terms in Eq 6 and 7 are not identifiable not only from factual outcome but also from any experimental data. (Line 235–236)
- For the same reason, the authors used an existing model not the data. What would be the implications of using a learned model? A learned model only reflects the factual outcome correctly and will estimate counterfactual harm incorrectly.
- It is unclear how benefit or harm are defined in medical or related research.

---

> ### Author Response · Authors · 2022-08-01
> **Response to reviewer rBDL**
>
> We thank the reviewer and look forward to continuing the discussion with them. Respectfully, we would like to raise a couple of concerns with the review.
>
> Firstly, the reviewer misresports the basic results of the paper (e.g. that we don't use a learned model for evaluation, that our results reduce to a trolley problem, etc), and we hope that in light of this the reviewer will consider revising their certainty score.
>
> Secondly, our paper is ranked as `good’ for soundness and presentation, and ‘fair’ for contribution, but then a high confidence rejection, which is due to its use of counterfactuals (due to a misinterpretation of the non-identifiability theorem) and equivalently its use of SCMs (which are also non-identifiable). Note that this grounds for rejection would apply to hundreds of papers that have been published at neurips and comparable venues. It appears the main reason for rejecting is because we use this established formalism, rather that due to the results themselves.
>
> Q1. It is unclear why we should define harm and benefit separately by taking max(0, difference in utility)
>
> This point is explained on line 191 immediately after the definition. It follows from the CCA definition of harm: harm (benefit) is the increase (decrease) in utility under the counterfactual action. The max splits the counterfactual utility distribution into positive (harmful) and negative (beneficial) components with the factual utility as origin.
>
> Q2. In the example in the introduction, I understand that why doctors would favor Treatment 1 vs 2. This feels so much like a trolley problem ... What’s your explanation? In HPU, U is -5, -1 and harm is -1. So with lambda 4, there is no difference between switching lanes?
>
> We are not sure how the reviewer has arrived at these figures. A worked example is clearly given in the text (full derivation in Appendix F). The expected utility is the same for both treatments. The harm is zero for treatment 1 and 0.1 for treatment 2. Therefore for any lambda > 0, treatment 1 is preferred. This is not equivalent to a trolley problem, as the factual outcomes for the two treatments are identical.
>
> Q3. Simply the P terms in Eq 6 and 7 are not identifiable not only from factual outcome but also from any experimental data. (Line 235–236)
>
> Non-identifiability is a well known result, but does not reduce the soundness or applicability of our results. Counterfactuals are necessary: As we show with the treatment example (now page 4), any factual measure of harm violates basic intuitive properties of harm. This example describes two treatments, one that is intuitively harmful and one that is not, but which have identical factual outcome statistics, i.e. P( Y = y | T = 1) = P(Y = y | T = 2). Any factual measure of harm is a function of these outcome distributions, and must assign the same harm values to T=1 and T = 2. Hence counterfactuals, and their required assumptions, are necessary when dealing with harm. The main definition of harm in ethics is also explicitly counterfactual. Secondly, good SCMs can be learned for real systems.  For example, “A ball is dropped and bounces X high. If it was dropped from twice the height, it would have bounced 2X high”. This is counterfactual is non-identifiable but is defined w.r.t a mechanistic causal model (the classical equations of motion). Clearly, this SCM can be learned. Finally, we point out that human ethical decision making uses counterfactuals, which must make the same assumptions. Making these formal and explicit in SCMs can only improve upon this.
>
> Q4. For the same reason, the authors used an existing model not the data. What would be the implications of using a learned model?
>
> This is not true. The model we use in section 6 was learned from real-world data (a meta-analysis for randomised control trials), as described in the text.
>
> Q5. It is unclear how benefit or harm are defined in medical or related research.
>
> We use the predominant definition of harm (the CCA) which is widely cited in the philosophy of medicine, e.g. [1]
>
> [1] Engelhardt, H. Tristram. "The concepts of health and disease."
>
> Q6. Def 1, why should noise be mutually independent?
>
> This is a standard assumption (causal sufficiency) used in the definition of SCMs, described in the cited materials. We have added an explanation
>
> Q7. Is utility fixed over the entire population? A is the agent's action but U is user's utility function. User != Agent. (Y would also be user's outcome...)
>
> We are not certain what population the reviewer is referring to. The setup is the same as in standard expected utility theory, and the utility function is an individuals (the user). We cant understand the rest of the question and ask for clarification.
>
> Q8. What's the practical importance of Y in defining harm and benefit when you have utility U? Y exists only to be marginalized out...
>
> standard in expected utility theory. Needed to describe outcome distributional shift independently of utility shift

---

> > ### Comment · Reviewer_rBDL · 2022-08-07
> > **Follow-up**
> >
> > Overall the authors answered kindly although my questions are somewhat vague.
> >
> > I mentioned Trolley problem because it involves "inaction" vs "action" of the agent, not just choosing "which lanes".
> >
> > I couldn't find added explanation about the Q6. Causal sufficiency (Markovian) is typically assumed in causal discovery to narrow the scope but not for general causal inference where an independent noise implies trivial identifiability. Even [7] does not mention independent noise. Researchers might model independent exogenous variables but each exogenous variable may affect multiple endogenous variables (semi-Markovian).
> >
> > Q7 was about whether U should be defined for each individual. Currently the utility function is fixed and individuals in the population (i.e., n data points) share the function but individuals may have different utility functions (and, hence, utility values).
> >
> > I still don't understand "SCMs can be learned for real systems." Maybe it's because the authors restricted to a Markovian case?
> >
> > My "initial" decision to recommend rejection is based on my counterfactual reasoning whether the scientific community would be benefit from the results in this paper. I liked the objective in its current counterfactual form. Results are sound and representation are good. Theoretical contribution seems fair. But, are the real-world application and complex domains free of unobserved confounders? Can you confidently say that this work "shows an immediate practical application with models currently in use in harm-sensitive applications."
> >
> > Anyway, I understand that my initial recommendation to reject is partly due to misunderstanding, and will raise to weak-accept. But I wish the authors recheck the assumptions and its implications.

---

> > > ### Author Response · Authors · 2022-08-08
> > > **Follow-up**
> > >
> > > We would like to really thank the reviewer for revising their score and for their helpful comments.
> > >
> > > Following the reviewers comments we have included a comment on line 145 and on line 69 to make it clear that we are assuming no unobserved confounders when deriving our theoretical results (sorry for missing this correction in our last update). We have included a discussion on this limitations due to this assumption in appendix K (paragraph beginning line 1228). We have also expanded appendix K to discuss the limitations of our work including those raised by the reviewer, and to include some examples of how our framework might be applied to some existing implementations.

---

### Official Review · Reviewer_jo7e · 2022-07-13

**Rating:** 7
**Confidence:** 4
**Ethics Flag:** Yes
**Soundness:** 3 good
**Presentation:** 3 good
**Contribution:** 3 good

**Summary:**

This paper uses a structural causal modeling (SCM) framework to define a statistical notion of harm which the authors name counterfactual harm (CH). The statistical definition extends a philosophical one named the comparative counterfactual account, which states that an action causes harm to a person if that person would have been better off had the action not occurred. The paper introduces a harm penalized utility (HPU) objective and compares the optimal actions according to this objective with others including an unpenalized utility and an objective that formalizes risk-aversion (penalizing the second moment) across several illustrative examples. The paper argues that avoiding harm requires causal reasoning, and specifically counterfactual reasoning, and shows that optimal actions according to other objectives can lead to causing harm. It presents several definitions and theorems to generalize this argument from the examples- the theorems show that various non-CH objectives can cause harm under a distribution shift in the data. The paper also illustrates its approach by comparison on a real data example involving dose responses for a drug.

**Questions:**

Q1: Is it really necessary to focus on counterfactuals to define an adequate notion of harm? What if we replace the counterfactual distribution in formula (6) with an interventional distribution? I believe the resulting definition would still be interesting and usable for many of the examples considered. An “interventional comparative account” of harm could say that an action will cause harm to a person if that person will (probably) be better off if a different action is chosen. Since many of the examples in the paper consider actions that may actually be feasible this definition may suffice for them. By contrast, the definition of counterfactual fairness (for example) considers attributes which cannot feasibly be changed, hence it is necessarily counterfactual and could not be interventional.

Q2: Is it too narrow to operationalize harm in a (simple) SCM framework? The model focuses on specific variables, while our intuition about harm usually involves a more open and wholistic idea of the world. Consider the two thieves example in Appendix D. The model specifies that Alice can only choose to rob or not rob Bob, while in the real world Alice could choose to simply warn Bob about Eve. Why should we apply the path-specific definition of harm that ignores an important way that Alice’s action can impact Bob’s wellbeing? I don’t think this resolves the paradox, but expanding the action space can. Similarly, a policy optimized to avoid harm according to a model built on one set of variables could cause harm involving other variables which were not included in the model. I think this is not just a curious theoretical issue because in algorithmic systems the model assumptions can be become built in and the data infrastructure can grow and increase in complexity, possibly cementing in place a harmful set of assumptions. However, the proposed narrow definition of harm would not address such issues.

Q3: Are the authors certain about the claims in the paper regarding the "first" statistical definition of harm? It would require a truly massive literature search to establish this. It seems to me the contribution of the paper is interesting enough to speak for itself, without needing to be claimed as a first in kind.

**Ethics Review Area:**

["Discrimination / Bias / Fairness Concerns", "Responsible Research Practice (e.g., IRB, documentation, research ethics)"]

**Limitations:**

Some already highlighted under the weaknesses, like reliance on untestable assumptions and use of unobservable constructs, and most importantly the first two questions listed above.

**Strengths And Weaknesses:**

The paper considers problems which are practically important. It is clear, interesting, and will likely generate a lot of discussion. A strength of its approach is the direct focus on things that truly matter, like utility, but a corresponding weakness is that much of the relevant quantities cannot be observed and require strong modeling assumptions, like the knowledge of an SCM. Note that this is not a specific weakness of the current paper, but one it shares with others that have been published in comparable conferences (including some of the cited related literature).

---

> ### Author Response · Authors · 2022-08-01
> **Reply to reviewer jo7e**
>
> We thank the reviewer for their thoughtful feedback and are encouraged that they describe our work as tackling a problem that is practically important, and describing the manuscript as clear, interesting, and likely to generate a lot of discussion.
>
> Q1 a: Is it really necessary to focus on counterfactuals to define an adequate notion of harm?
>
> To answer this question we reiterate a key example that was perhaps not emphasised enough in the text (Example 1 the introduction & appendix F), where we show that any measure of harm that is based on factual inference (and so doesnt make these assumptions) violates basic intuitions about harm (not being able to tell the difference between clearly harmful and clearly non-harmful actions). From this we argue that as these assumptions are necessary for dealing with harm, they aren’t weakness of our specific definition but of any definition that can satisfy our basic intuitions about harm.
>
> The example describes two treatments, one that is intuitively harmful and one that is not harmful, but which have identical factual outcome statistics, i.e. P( Y = y | T = 1) = P(Y = y | T = 2). Any factual measure of harm is a function of these outcome distributions, and must assign the same harm values to both. Of course, one could choose an objective function J(t, y) that depends on T, and implicitly favours T = 1. But then one just has to observe that under the distributional shift where the causal mechanisms for Y are swapped f_Y(T = 1, e_Y) <--> f_Y(T = 2, e_Y), this utility function will favour T = 1 which is now the harmful treatment. So even before we get to the predominant definition of harm (the CCA, which is explicitly counterfactual) we see that any factual measure will not work.
>
> Q1 b: What if we replace the counterfactual distribution in formula (6) with an interventional distribution?
>
> To see what happens if we replace the counterfactual formula in (6) with an interventional distribution, consider the case where U(a, x, y) = U(y) and there are two outcomes Y=0 and Y = 1, where Y = 1 has the higher utility. Replacing P in (6) with its interventional equivalent P(y_a | x) instead, i.e. the factual outcome distribution, reduces the harm to P(Y_a = 1 | X = x)(U(Y = 1) - U(Y = 0)) which is the casual effect of do(A = a) times the utility difference. So the harm would reduce to a treatment effect, which as we have shown in the above example is not suitable for capturing harm (e.g. it cannot differentiate between treatments 1 and 2 in the motivating example).
>
> Q2 a: Is it too narrow to operationalize harm in a (simple) SCM framework? [+ question about the two thieves example]
>
> We use the SCM framework to derive our theoretical results and definitions as is standard in Pearlean causality (e.g. all the main results of the do-calculus are derived in the SCM framework), and doesnt limit our theoretical results to simple systems. For practical applications, there are lots of recent works (including those published at neurips) that extend SCMs to complex domains (for example, deep structural causal models, which have been applied to medical imaging).  We discuss these in relation to our work in a new appendix K.
>
> In the two theives example, if Alice’s default action was to warn Eve then the harm value would be higher. So to determine harm we need to have a model that incorporates the desired default action. That models with oversimple action spaces may result in bad decisions is true  for any model-based methods, not just our result—It would also be true for the proposed `interventional harm’. For example, treatment effects cannot be used to determine the best treatment if we miss important available treatments from the analysis.
>
> Q2 b: Similarly, a policy optimised to avoid harm according to a model built on one set of variables could cause harm involving other variables which were not included in the model....however, the proposed narrow definition of harm would not address such issues.
>
> This can be applied to all methods that use statistical modelling in decision theory. When a doctor makes a clinical decision based on a counterfactual inference (i.e `the patient would have responded better to a different treatment’), they are necessarily basing this on a mental model that includes untestable mechanistic assumptions (in order to support counterfactual inferences). This is an inescapable fact, but clearly they are capable of doing it (good inductive biases and heuristics for learning SCMs exist), and while it is possible that causal models make mistakes due to wrong assumptions and oversimplifications, the same is true for these mental models that support a huge number of human decisions in medicine, law, etc. We believe that making these models formal and explicit is a route to reducing the effect of these errors rather than increasing them.

---

> > ### Comment · Reviewer_jo7e · 2022-08-08
> > **Follow-up**
> >
> > About non-counterfactual definitions, the reply has not really addressed my point.
> >
> > Consider this example: I have been asked to provide an ethics audit for a company which is building a doomsday machine that will destroy the universe when someone presses the activate button A = 1. This technology has never existed before, so there is no historical experience for us to consider any counterfactuals. Clearly the machine will do harm if activated, and clearly an interventional definition of harm would capture that.
> >
> > In the reply to Q1b, you provide a counter-example based on a strange utility function. That example shows an interventional definition of harm would not be appropriate for all possible examples. I have just given an example which shows a counterfactual definition also would not apply to all possible examples. My point is not to push the authors to change the proposed definition in this paper, but just to see that it is a specific choice to focus on counterfactual reasoning. The paper could more clearly communicate how this choice is specific and has limitations. It could, for example, mention that SCMs with an interventional definition of harm may be more appropriate for some examples.
> >
> > My previous comment about acknowledging/communicating limitations also applies to Q2. Again, I am not asking the authors to change the proposal. Formal models, like SCMs or any mathematical models, are precise, narrow, technical, rigorous, etc, but also always wrong in some important ways because the real world is open ended and does not have clear boundaries and definitions of variables, etc. If we take the open ended principle to "do no harm" and make it formal with an SCM that implies Alice should rob Bob, that shows the formalization process went wrong somewhere. Any reasonable human who hasn't been indoctrinated with mathematics would resolve that example by saying Alice should warn Bob about Eve (and also not be out looking to rob anyone in the first place). The lesson of this example, in my opinion, is not that some path specific definition in the same SCM resolves the example, but a reminder that any formal model (causal or not) could fail to capture something important about what people mean by "do no harm."
> >
> > Even with no formal model, I think a less abstract example would make the point more clear (instead of one about Alices and Bobs). I have been hired by a giant tech monopoly or superpower government to make its algorithmic systems less harmful. It is designing a system to automatically steal money from everyone, and it wants to A/B test whether to steal 100 currency units or 101. Those are the only options within the technical specification. Since I am human, I can make choices outside of a technical specification. My professional choices are to (A) proceed to write down an SCM and formally prove that stealing $100 does less harm, report this to my manager, get rewards and accolades and advance in my career, or (B) be an actually moral actor and challenge the technical specification, cause problems for my manager, possibly risk my career advancement, etc. Again, I think any reasonable understanding of "harm" that hasn't already been restricted to a mathematical model would not struggle with which action is morally correct. And because the example is no longer abstract but includes a dimension of professional advancement it also illustrates the danger of having formal models that we can use to rationalize morally objectionable systems.
> >
> > To reiterate in conclusion: I know this is not a limitation of this specific paper. I do not expect the authors to provide a complete rigorous methodology that does not have any of the shortcomings that all other rigorous methods suffer from. All I would expect is some clear acknowledgement and communication about limitations.

---

> > > ### Author Response · Authors · 2022-08-08
> > > **Follow-up**
> > >
> > > Thank you for clearing this up and apologies for misunderstanding your point. In our previous rebuttal we interpreted your comment as saying that this was a problem with our approach specifically, rather than this is a problem with all algorithmic approaches to making morally relevant decisions, and that it is precisely for this reason that the issue should be discussed in the paper. This was an error in communication on our part, and we have included a discussion on the limitations of our approach in appendix K, including (in the first section) a discussion clearly stating that our results do not resolve the issues you raise.
> > >
> > > We totally agree with you on these points. In fact the entire motivation behind writing this paper was to point out another yet-unseen failure mode for any algorithmic decisions (not using counterfactual reasoning, as humans do, to determine harm). This we think is an important point, because no matter how good models become (even if they are better than humans—e.g. the AI knows that it can warn Bob in the market, but Alice does not), then it will still make mistakes that no reasonable human would. There is a tension where presenting a solution to one problem can be misconstrued as a solution to all problems in algorithmic harm (model misspecification, robustness, etc)—we hope it is now clear in the paper that this is not the case.
> > >
> > > Next we will address the world-ending machine example.
> > >
> > > Short answer: The example you give is not a counterexample to the counterfactual definition. In this example the counterfactual harm reduces to the treatment effect as the outcomes are deterministic. So the counterfactual harm is equivalent to the proposed causal measure of harm. Also it is not correct that counterfactual analysis requires more `historical experience' than interventional analysis, they are both defined over the same action-outcome space (A, X, Y).
> > >
> > > Long answer: First we note that making rung 3 counterfactual inferences requires no more or less access to historical examples as making rung 2 causal inferences. For example, if we wanted to estimate the causal effect from data we would have to push the button multiple times. As we cant do this we have to ask where the inference “if I press the button A = 1 the world will end” comes from. As this is not a repeatable experiment the inference must come from a predictive model (e.g. we know enough about how the device works and how the universe works to know that the device will destroy the universe). So whatever case you choose (expert reasoning, causal inference, counterfactual inference), we have to assume there is a model. Given this model we calculate the expected counterfactual harm as follows—given that I press this button, sum over all factual outcomes that can occur (in this case, the world ends with certainty) and for each of these compare the factual utility to the counterfactual outcome that would occur if I did not press the button (the world not ending with certainty). Because of the deterministic outcomes, the counterfactual harm is just the difference in expected utility given do(A = 1) and do(A = 0). So you see that the counterfactual definition of harm works perfectly well in this setting. We can also describe what happens for indeterministic outcomes but it is similar to the treatment example.
> > >
> > > Re: Q1b, the utility function U(y) is just taken because its simple and intuitive (the users prefer living over dying, and have no intrinsic preference for one treatment over the other). Its easy to come up with equivalent counterexamples for any U(a, y) (this is what we do in section 5), but we kept it simple here to try and point out that in this intuitive setting (people like living, dont like dying, two treatments) we can see the factual approaches cannot work for all cases. One of the the main results of the paper is a proof that rung 2 or 1 definitions of harm won’t work, so we disagree that a counterfactual definition is a choice on equal footing with a causal definition.
> > >
> > > Also note that if there was no counterfactual measure that robustly captures harm, there would be no interventional measure, as counterfactuals refine interventional distributions. So an equivalent proof in the opposite direction is impossible. To see the relation between counterfactual harm and any intervention definitions, note that the counterfactual harm reduces to the difference in expected utility under different interventions if you make the additional assumptions;  `counterfactual independence’, i..e that P(y^*_{\bar a}, y_a) = P(y^*_{\bar a})P(y_a), or deterministic outcomes being two examples. We can state this in the text if you think it would be useful to the reader.
> > >
> > > We also just want to clarify that the path specific harm is not intended to resolve model misspecification problems but to point out that in some cases what we mean when we talk about harm is a path specific variant of harm.

---

> > > > ### Comment · Reviewer_jo7e · 2022-08-09
> > > > **Counterfactuals are (still) not always necessary**
> > > >
> > > > The world ending example being deterministic does not resolve the issue with the example. Interventions, counterfactuals, and harm are all concepts which are not necessarily probabilistic. What this example shows is that we do not need an SCM and counterfactual reasoning in order to choose the less harmful course of action. So, again, a counterfactual definition is not always necessary.
> > > >
> > > > > we can see the factual approaches cannot work for all cases
> > > >
> > > > No approach can work for all cases. No free lunch, all models are wrong, etc.
> > > >
> > > > > One of the the main results of the paper is a proof that rung 2 or 1 definitions of harm won’t work
> > > >
> > > > If this said "won't work for some broad class of examples and under some additional assumptions" then I would agree. There are cases where a counterfactual approach seems necessary and clearly the best, hence the value of this paper and the reason I did not choose any rating like "reject." But I do not agree that a counterfactual definition is always necessary or always the best. The lesson of Pearl's ladder is not that "higher is always necessary or better" or "lower is always wrong." Someone who cares more about the epistemological cost of making assumptions could just as well invert the ladder and say that counterfactual reasoning is the lowest, worst form of reasoning because it relies most strongly on assumptions.
> > > >
> > > > Also again: I am not requesting any significant change in the paper. When I initially asked Q1 in my review, I was not asking for the paper to be re-written about causal harm in general. It is fine to focus on rung 3 and argue for its necessity in some contexts as this paper does. I just think it would be useful context for a reader to know that an interventional definition of harm could be pursued and may be good enough for other examples, even though that is not the choice of the current paper. And I think this context is important given this paper argues against an interventional definition in some contexts, hence some readers may mistakenly conclude an interventional definition would always be wrong.

---

> > > > > ### Author Response · Authors · 2022-08-09
> > > > > **reply**
> > > > >
> > > > > I think this misunderstanding is due to us using a single world differently. One important point that I hope we can agree on is the distinction between a definition and a method. A formal definition must work in all scenarios, otherwise it has to be discarded. Methods dont have to work in all scenarios and in general cant (no free lunch theorems etc). For example a definition of the treatment effect must match our intuitions for how a treatment effect behaves in all situations (e.g. they are zero if there is no directed path from X to Y), and no associative definition of the treatment effect robustly capture the treatment effect. But there are associative methods for estimating treatment effects that work in some situations. In your comments you refer to a causal definition of harm, and Im going to assume you mean a definition as described here. If what you meant was that rung 3 inference is not always necessary to measure harm, then I agree and please skip to the last paragraph.
> > > > >
> > > > > Assuming you mean definition...
> > > > >
> > > > > Running analogy. Imagine our paper rather than defining harm was defining the treatment effect, in terms of distributions P(y | do (x)). We show that any associative definition cant work, by using the obvious example where X and Y are correlated but purely by a common cause, so while P(y | x) \neq P(y | x’), the treatment effect is zero for all x, e.g. P(y | do(x)) = P(y | do(x’)). This is a “weird example” because in the general case there will be both direct and common causes between X and Y. Nevertheless, it is sufficient to show that there can be no definition of causal effect that is purely in terms of associative statistics that matches our intuitions.
> > > > >
> > > > > “I do not agree that a counterfactual definition is always necessary or always the best”.
> > > > >
> > > > > You claim this based on an example where the counterfactual harm reduces to an interventional measure (if you calculate equation (6) in our paper for your example, you get H(A = 1) = E[U| do(A = 0)] - E[U| do(A = 1)], which matches our intutions, see end of rebuttal). Consider the analogous situation with defining the treatment effect. There are some scenarios where causal effects reduce to associative distributions P(Y | do(x)) = P(Y | x) (i.e. no confounders). This does not mean that the causal definition of the treatment effect is not always best, or “not necessary”---definitions are definitions and must work independently of the situation. There are situations (e.g. pure confounders) where any associative definition of the treatment effect will fail. The same is true for rung 2 definitions of harm as we prove by counterexample.
> > > > >
> > > > > “What this example shows is that we do not need an SCM and counterfactual reasoning in order to choose the less harmful course of action”
> > > > >
> > > > > The same is true in the treatment effect analogy. In some situations, we dont need interventions to measure treatment effects. Any attempt to maximise the treatment effect by maximising P(y | x) rather than P(y | do (x)) is guaranteed to fail in some environment (e.g. those where X and Y are only related by a common cause).
> > > > >
> > > > > “it would be useful context for a reader to know that an interventional definition of harm could be pursued and may be good enough for other examples”.
> > > > >
> > > > > Our paper proves by counterexample that a purely interventional definition of harm is not possible (definition being the main word), and claiming this in our paper would contradict all our main proofs. But again we think what you mean here is causal methods for estimating harm which we discuss now.
> > > > >
> > > > > Last paragraph... (assuming you mean method)
> > > > >
> > > > > We agree that counterfactual inference will not always be necessary to measure harm, as there as some situations where the CCH can be evaluated using rung 2 inference (as you have described). This is explicitly explored in section 5 where we consider interventional objective functions that are harm averse in a single environment. But the point of section 5 is to show that any such measure is not robust to the situation changing, and there is always a situation where they fail (analogous to associative measures for treatment effects). For example, see the third paragraph above). We agree that simply evaluating the full counterfactual definition isnt the best method for estimating harm in all cases (e.g. when the SCM isnt known). What we will do is include a note in the paper describing
> > > > >
> > > > > 1. When does the CCH definition reduce to a rung 2 definition
> > > > > 2. What are rung 2 measures that tightly bound the CCA in all situations (basically allowing one ensure harm aversion without needing and SCM)
> > > > >
> > > > > Is this what you are requesting? Just because time is tight we include a calculation of counterfactual harm in your example, just incase this is still a sticking point
> > > > >
> > > > > Y = 0, world end, Y = 1 work doesnt end. Y = 1 - A (world ends if we push the button, doesnt end if we dont). P(Y_{A = 0} = 1 | Y_{A =1} = 0) = 1. So H(A = 1) = U(A = 0, Y = 1) - U(A = 1, Y = 0).

---

> > > > > > ### Comment · Reviewer_jo7e · 2022-08-09
> > > > > > **It's not about definitions vs methods**
> > > > > >
> > > > > > > A definition must work in all scenarios, otherwise it has to be discarded
> > > > > >
> > > > > > I'm not sure how to make sense of this, since for example there are scenarios where the concept of harm does not even apply. Distinctions between definitions, methods, theorems, etc, are only conventions of professional writing in certain disciplines, that is not what my objection is about.
> > > > > >
> > > > > > There are an abundance of purely associative definitions of fairness in the literature. There are many causal examples showing how these associative definitions can be "wrong" under some scenarios. But I have never seen anyone arguing that all associative definitions have to be discarded because they don't "work in all scenarios."
> > > > > >
> > > > > > > Any attempt to maximise the treatment effect by maximising P(y | x) rather than P(y | do (x)) is guaranteed to fail in some environment
> > > > > >
> > > > > > Yes, but nobody has yet banned all health research (for example) that is based on associations, or argued that we cannot even *define* "risk factors."
> > > > > >
> > > > > > Now if we just take this argument one more step up the ladder we reach the current disagreement. You have given a class of examples where an interventional definition would be "wrong," but this does not mean it's impossible to propose or use such a definition in other scenarios.
> > > > > >
> > > > > > Let me reiterate a point in my initial review to contrast with the definition of counterfactual fairness. In that setting, since a person's sensitive attribute (e.g. race, sex) is already determined, it is *necessarily counterfactual* to consider what their other variable values *would have been* if their sensitive attribute had been different. An interventional definition in that setting would necessarily have to focus on some shallow/minimal intervention like manipulating the perception of race based on changing names on a CV, without actually modeling how else that person's entire life (and hence CV) might also have been different in a counterfactual world where they had a different race (and not just a different name on their CV). By contrast, when we are considering harm we can be thinking about examples that are entirely based in the future (like whether to build a doomsday machine), where the facts have yet to be determined and hence the reasoning involved is not necessarily counterfactual.
> > > > > >
> > > > > > > Our paper proves by counterexample that a purely interventional definition of harm is not possible
> > > > > >
> > > > > > Hence I must continue to disagree with the above statement.

---

> > > > > > > ### Author Response · Authors · 2022-08-09
> > > > > > > **Reply**
> > > > > > >
> > > > > > > After some discussion we agree with your points and have tried our best to incorporate them in our paper. Will the following changes be sufficient?
> > > > > > >
> > > > > > > 1. In all claims in the paper of the form `we show that no associative measure of harm can work', we will change these to `no associative measure of harm can satisfy the CCA definition' or `no associative definition of harm can satisfy our intuitions in the treatment example'. Which is better than us saying / implying `no other definition can work in all cases'. We rewrite the abstract, introduction section 5 and conclusion to reflect these changes.
> > > > > > >
> > > > > > > 2. We have included a section in the appendix K (K.3. limitations) where we describe situations where using counterfactual inference to determine harm is not the best option, and given an example of a factual upper bound to the counterfactual harm so that the CCA can be used in these situations to ensure strict harm aversion even when counterfactual inference is not possible.
> > > > > > >
> > > > > > > 3. We will expand our discussion of other definitions of harm in Appendix B (currently these other definitions are cited in the introduction but not expanded on beyond their criticisms of the CCH).
> > > > > > >
> > > > > > > Thanks for your comments!

---

> > > > > > > > ### Comment · Reviewer_jo7e · 2022-08-09
> > > > > > > > **Nice**
> > > > > > > >
> > > > > > > > I think these are good changes that improve the paper, and I have updated my rating accordingly

---

> > > > > > > > > ### Author Response · Authors · 2022-08-09
> > > > > > > > > **Update details + thank you**
> > > > > > > > >
> > > > > > > > > Massive thank you for taking the time to discuss this with us and improve our paper! Just to update you on the specific changes we made.
> > > > > > > > >
> > > > > > > > > Abstract updated, making it clear that we are translating a specific definition of harm, and that our results re: factual approaches apply to specific cases.
> > > > > > > > >
> > > > > > > > > Line 132-133 updated to reference further discussion of other definitions
> > > > > > > > >
> > > > > > > > > Line 328 updated to make it clear that these policies are harmful with respect to the CCA definition of harm
> > > > > > > > >
> > > > > > > > > Line 342 made it clear that these needlessly harmful actions are w.r.t Definition 5, which refers to the CCA definition
> > > > > > > > >
> > > > > > > > > Conclusion: toned down the language to `are unable to avoid harmful actions in certain situations’. Refer to factual bounds on harm, limitations and related works (all of which are discussed in appendix K)
> > > > > > > > >
> > > > > > > > > Appendix B. Included discussion of two other definitions of harm
> > > > > > > > >
> > > > > > > > > Appendix K. A section discussing limitations of our approach including using counterfactual inference for measuring harm, and discuss bounding counterfactual harm using factual distributions.

---

### Review · Ethics_Reviewer_Rk7s · 2022-07-28

**Recommendation:**

It would help the readers to discuss the reasons for choosing the CCA approach instead of other normative frameworks because, in the current state, it seems that the choice has been given by the simplicity of adapting it to statistics instead of a reasoned and justified ethics choice. It's important not to underestimate the concept of "harm" in AI, and to argue and justify using an ethical framework instead of another. The concept of "harm" could also use other references in its definition (in both ethics and ML literature).

**Ethics Review:**

This paper doesn't raise direct ethical issues as listed in the NeurIPS ethical guidelines, hence the "no" above. However, the statistical approach used for a definition of "harm" has its limits when combined with the philosophical one of the CCA approach, which is already controversial in the field (eg see: Carlson, 2019). Moreover, I am afraid the authors do not clearly distinguish between questions of bioethics (like in the examples and appendix), where the notion of "harm" is more easily defined, and questions of AI ethics/machine ethics and their complexity. The definitions given in the introduction about "harm" confuse ethics and law and have a somewhat superficial ground. The real-life applications of this method could be ethically sensitive, which seem to work on a logical level but would be hard to imagine applied to AI.

---

### Review · Ethics_Reviewer_2KKT · 2022-08-08

**Recommendation:**

The authors can discuss the simplification of the action space as limitation in the paper, and provide recommendation on real life scenarios where the agents could face multiple actions/ decisions and how they should adapt the SCM framework in those scenarios.

**Ethical Issues:**

Yes

**Ethics Review:**

This paper proposes a statistical definition of harm called counterfactual harm and creates a structural causal models framework that incorporates the harm into algorithmic decisions.
I see two potential ethical concerns related to the paper: 1. over-simplifying the actions of the agents to binary and 2. potentially unable to be applied to all situations.

Regarding the two thieves example, on top of either robbing or not robbing Bob, Alice does have the option of warning Bob, calling the police, stopping Eve etc. These additional non-counterfactual actions have the potential to have higher utility, and thus making Alice robbing Bob not the most morally relevant action, but merely better than doing nothing (not robbing Bob). I agree that using such counterfactual harm SCM framework could potentially mislead agents into making non-optimal decisions when there could be multiple actions or if counterfactual is unknown.

Without undermining the contribution, the paper can discuss the simplification of the action space/ limitation, that the SCM framework only evaluate the least harmful action in relation to the single counterfactual, and that the model is more applicable to scenarios where the agents’ actions are binary and counterfactuals are known. When the agents have multiple possible actions, they should perhaps repeatedly evaluate each default action with its counterfactual. Overall I see this could be addressed by the authors by a relevant discussion in the paper/ appendix.

---

### Comment · Area_Chair_2JCk · 2022-08-07
**Discussion with Authors**

Dear Reviewers! Thank you so much for your time on this paper so far.

The authors have written a detailed response to your concerns. How does this change your review?

Please engage with the authors in the way that you would like reviewers to engage your submitted papers: critically and open to changing your mind. Thank you Reviewer rBDL for your initial engagement!

Looking forward to the discussion!

---

### Meta-Review · Area_Chair_2JCk · 2022-08-26

**Recommendation:** Accept
**Confidence:** Certain

**Metareview:**

All reviewers agreed that this paper should be accepted because of the strong author response during the rebuttal phase. Specifically the reviewers appreciated the motivation of the paper, its clarity, and the author clarification of the method, its assumptions, and scope during the rebuttal. Authors: please carefully revise the manuscript based on the suggestions by the reviewers: they made many careful suggestions to improve the work and stressed that the paper should only be accepted once these changes are implemented. Once these are done the paper will be a nice addition to the conference!

**Award:**

No

---

### Decision · Program_Chairs · 2022-09-14

Accept